# A 2700-yr record of Cascadia megathrust and crustal/slab earthquakes from Acorn Woman Lakes, Oregon

Ann E. Morey[1] and Chris Goldfinger[2]

[1]Cascadia Paleo Investigations, Corvallis, OR, 97330, United States

[2]College of Earth, Ocean and Atmospheric Sciences, Oregon State University, Corvallis, OR, 97331, United States

*Correspondence to*: Ann E. Morey (ann@cascadiapaleo.org)

**Abstract.** We infer a ~2,700-year history of Cascadia megathrust and other earthquakes from two small mountain lakes located 100 km inland of the coast near the California/Oregon border. We use the characteristics of a disturbance deposit in the historic portion of the sediment cores attributed to the 1700 CE Cascadia earthquake to identify Cascadia earthquake deposits downcore. This deposit is composed of light-coloured silt sourced from the delta, has extended organic grading of the deposit tail, and a basal contact with evidence of rapid loading. Eight deposits downcore have the characteristics of this deposit. An age-depth model suggests that six of these are temporal correlatives to the largest margin-wide marine turbidite event deposits from Goldfinger et al., 2012, (deposits T1 through T6), whereas the two deposits with some of the characteristics are potential correlatives of smaller deposits T5a and T5b. We use the characteristics of the 1873 CE Brookings earthquake suggesting subaerial landslides attributed to a crustal fault earthquake, to identify similar deposits downcore. As a result, temporal correlatives of T2a and T3a and other smaller deposits from the marine record were identified as likely crustal fault earthquakes. These results suggest that small Cascadia landslide-dammed lakes from distances of 100 km inland of the coast with sufficient sedimentation rates (~1-2 cm/decade) and mixed clastic and organic content may be good recorders of subduction earthquakes. Furthermore, southern Cascadia crustal earthquakes likely partially explain the more frequent earthquakes in southern Cascadia and suggest a previously unrecognized hazard in the region.

## 1 Introduction

The hazard posed by Cascadia earthquakes is a function of earthquake size, location, and frequency (Petersen et al., 2020; Priest et al., 2009; Goldfinger et al., 2012; Wang et al., 2013). Evidence of earthquake magnitude comes from estimates of turbidite thickness of offshore seismogenic turbidites (Goldfinger et al., 2012), tsunami size and deposit distribution in coastal marshes in lakes (Kelsey et al., 2005; Witter et al., 2012), and evidence from the spatial pattern and amount of coseismic subsidence (Atwater and Hemphill-Haley 1997; Kelsey et al., 2002, Nelson et al., 2008; Witter et al., 2003; Graehl et al., 2014; Kemp et al., 2018; Nelson et al., 2020).

The most recent research suggests that southern Cascadia has experienced more frequent and variable earthquakes compared to full-margin ruptures (Nelson et al., 2006; Goldfinger et al., 2012; 2013; Morey et al., 2013; Priest et al., 2017; Goldfinger

et al., 2017; Milker et al., 2016), however there are uncertainties in the timing and number of events represented in the different records. This uncertainty arises from the location, depositional environment, type of record and dating accuracy and methods used to develop earthquake chronologies. The question of recurrence intervals in southern Cascadia is particularly important because the National Seismic Hazard maps (Petersen et al., 2020) currently rely on the limited data that exists for southern Cascadia, primarily the offshore record of marine seismogenic turbidites (Goldfinger et al., 2012). These records suggest that

the southern Cascadia recurrence interval is half that of full margin ruptures (averaged over the Holocene). The tsunami record from Bradley Lake, Oregon (southern Cascadia; Kelsey et al., 2005; Witter et al., 2012; Priest et al., 2017), with an intermediate recurrence interval of ~390 years, also has fewer tsunamis over a shorter period of time.

Morey et al., 2024 (this volume), suggest that sediment disturbed by earthquakes in Lower Acorn Woman Lake produce deposits that are different from those produced by other types of disturbances, such as floods, and that there are differences

among deposits formed in response to subduction earthquakes from those deposited in response to other types of earthquakes. Here we use this information to address the hypothesis that deposit J, attributed to the 1700 CE Cascadia earthquake in Morey et al., 2024 (this volume), was formed in response to shaking from a megathrust earthquake by comparing the frequency and timing of older deposits with similar characteristics to the frequency and timing of published records of pre-1700 Cascadia earthquakes from Goldfinger et al. (2012). A result of similar timing and frequency of disturbance deposits in the Acorn

Woman Lakes with other types of records of Cascadia megathrust earthquakes would be strong evidence that the sedimentary record from Acorn Woman Lakes, Oregon, record megathrust earthquakes.

## 2 Methods

### 2.1 Setting

Upper and Lower Acorn Woman Lakes (42°01′55″ N, 123°00′56″ W), previously called Upper and Lower Squaw Lakes, are

located in Klamath-Siskiyou Mountains, ~180 km inland of the trench (Figure 1) at the latitude of the boundary between the Juan de Fuca and Gorda Plates. The lakes were formed when a landslide dammed Acorn Woman and Slickear Creeks near their confluence (Figure 2). Both lakes have large subaerial deltas. The lower lake has two watersheds, one draining bedrock containing potassium-rich schist rocks (Acorn Woman Creek watershed) and the other draining bedrock containing calcium-rich rocks, including amphibolites (Slickear Creek watershed). Details can be found in the companion paper, Morey et al.,

2024 (this volume).

We collected Lower Acorn Woman Lake sediment cores during the summers of 2013, 2014, and 2015. We used a modified Livingstone corer (Wright, 1967), deployed from a custom platform fitted with a stainless-steel pipe attached to two inflatable rafts (2013) or canoes (2014), to collect cores SQB 1, 2, 4,5, 6, 7, and the surface sample (ss) for cores 1 and 2. A Kullenberg piston corer (Kelts et al., 1986) was used to collect cores in 2015. Surface samples from the same locations were collected with

a gravity corer. The Kullenberg and gravity coring devices were deployed from a stainless-steel platform attached to a large

pontoon-style raft (LacCore; 2015). We acquired single-beam bathymetric data in May 2015 by canoe fitted with a Garmin GPS-enabled "fish finder" and receiver (1500 m/s).

We described the sedimentology and deposit characteristics using the following data types: colour (Munsell chart), sediment texture, composition, grading, and contact characteristics as described in Morey et al., 2024 (this volume). Particle-size distribution data (volumetric % by size) were determined by laser diffraction analysis using a Horiba Grain Size Analyzer (LA-920; LacCore) or Beckman Coulter Grain Size Analyzer (LS 13-320; Oregon State University) after organic matter was removed using a 30% hydrogen peroxide solution at 85°C (overnight). Magnetic susceptibility (volumetric; $k$) was measured using a Bartington MS2E point sensor at 0.5 cm resolution. We acquired combustion data at 0.5–1.0 cm intervals through disturbance deposits and less frequently between these disturbance deposits (cores SQB2 and SQB14 only), resulting in data for the percentage of inorganic content (clastic particles other than $CaCO_3$), percentage of organic matter (degraded and particulate plant material), and percentage of $CaCO_3$ (calculated from dry weights). We acquired radiodensity (computed tomography, CT) data using a Toshiba Aquillon 64 slice CT unit at the Oregon State University Veterinarian Hospital (at 0.5 mm resolution).

We identified disturbance deposits in cores as abrupt increases in radiodensity, as described in Morey et al. (2013). These disturbance deposits were then correlated throughout Lower Acorn Woman Lake using radiocarbon ages from detrital macrofossils and physical property data using modified well-log techniques (Fukuma, 1998; Karlin et al., 2004; Abdeldayem et al., 2004; Hagstrum et al., 2004; Waldmann et al., 2011; Goldfinger et al., 2012; Patton et al., 2015. These sediment physical property data allow deposits to be correlated based on deposit composition and structure (Amy & Talling, 2006) and is a widely used method to correlate seismogenic marine and lacustrine turbidites.

We sampled the Lower Acorn Woman Lake cores for radiocarbon after splitting the cores longitudinally. Fragile detrital plant macrofossils sampled from targeted horizons of undisturbed sediment, were cleaned and dried, then analysed by AMS (accelerator mass spectrometer) for radiocarbon. We selected the target horizons for sampling based on a tentative relationship between the dated sequence from Upper Acorn Woman Lake (Colombaroli & Gavin, 2010) and the Lower Acorn Woman Lake stratigraphy. We did not acquire [210]Pb and [137]Cs data for Lower Acorn Woman Lake cores to get historic sedimentation rate data because the upper portions of the sediment cores contain two thick clastic units (found lake-wide and of varying thickness) with evidence of erosion at the basal contact, which violates the assumptions of the dating method of continuous sedimentation required to create a sedimentation rate curve ([210]Pb) or there was too much sediment missing below the horizon of interest ([137]Cs; below deposits A and B). We used the correlation of physical property data, in particular magnetic susceptibility and radiodensity, between the upper and lower lakes to infer that the younger of these disturbances, deposit B, settled in 1964 (as demonstrated in the supplementary data).

An age-depth model for the historic portion of the record was developed from an event-free sequence (e.g., Enkin et al., 2013; Hamilton et al., 2015; Goldfinger et al., 2017) using radiodensity. The base of each disturbance deposit was determined to be the location where radiodensity rapidly increases from background sediment and the top of the deposit was determined to be where radiodensity drops below background levels. Disturbance deposits without evidence of inter-event sedimentation are

treated as a single disturbance deposit for the purpose of the age model. Disturbance deposits show significant variability downcore, which complicates the boundary identifications (described in more detail below). A final age-depth model was created using a P_sequence in the Bayesian software OxCal (Bronk-Ramsey, 2017).

## 2.2 Inferred characteristics for earthquake types

Evidence presented in Morey et al., 2024 (this volume), suggests that there are two primary types of earthquake-triggered
disturbance deposits, and that both are different from the deposits of other types of disturbances (i.e., flood deposits) in the historic portion of the record from Lower Acorn Woman Lake. The two primary types of earthquake-generated disturbance deposits identified and described in Morey et al., 2024 (this volume) are presented below.

Type 1 (Figure 3a). This deposit type (represented by deposit J) shows evidence of a possible bypass turbidite at the base followed by a light coloured (2.5Y 4/1), well-sorted medium silt sourced from the Slickear Creek watershed. The silt in deposit
J is a thick (~7-15 cm), dense (~1,000 HU at the base), weakly graded, medium to fine-grained silt unit with an organic-rich tail. The base of the silt is composed of fine-grained, well-sorted silt (~90% inorganics) that appears "clean" (lacking other components such as broken diatoms and organic matter). The basal silt is 1.5-4.0 cm thick (depending on location in the lake) and becomes less-well-sorted upward with grading. As grading proceeds upward, the silt becomes more fine-grained, and the organic content gradually increases upward. The particle-size distribution at the base of the deposit is narrower than the rest
of the disturbance (as shown in Figures 3a and 3b) and pure (predominantly silt with only trace amounts of diatoms and organic particles). This fine-grained (medium-fine silt) sediment was interpreted to sink into the less-dense sediment below, which we suggest may be the result of loading ("LOAD" in Figure 3a), although these structures may be influenced by ground motions after deposition. X-ray diffraction (XRD; see Morey et al., 2024; this volume) demonstrates that this silt is composed primarily of sediment sourced from the Slickear Creek watershed. Above the deposit tail, background sediment contains a different suite
of diatoms compared to prior to the earthquake after the tail deposit suggesting a post-earthquake change in community structure (the types and relative abundance of species living in the water column). It was inferred that the processes triggered by the earthquake removed organisms during settling, altering the types of organisms present in the water column. Deposit characteristics and timing suggest that deposit J was formed in response to the 1700 CE Cascadia earthquake, therefore Type 1 deposits are inferred to be the result of Cascadia earthquakes.

Type 2 (Figure 3b). This type is a complex sequence represented by deposits H and I. Deposit I is a turbidite composed of dark grey (GLEY2 4/5PB) disaggregated schist-derived silt with visible mica fragments. It displays reversed, then normal grading from a coarse- to medium-grained silt upward to form a short organic tail followed by a thin layer of deciduous leaves (in some cores, forming the boundary between the schist turbidite and the silt from deposit H above). In contrast, deposit H is a light coloured (2.5Y 4/1 at the base) Slickear Creek watershed-sourced silt in core SQB2 and SQB5 (shallower water sites). Grading
proceeds from poorly sorted medium silt upward to a more well-sorted medium silt, and loss on ignition data indicate an upward increase in the ratio of organics to inorganics with grading. The event deposit in SQB2 appears to have a tail which is hummocky with respect to radiodensity instead of smoothly grading upward. This hummocky nature correlates to deep-water

turbidites in core SQB9 which are interpreted to be the result of a crustal earthquake sequence with a possible subduction influence (see Morey et al., 2024, this volume). The interpretation, based on deposit characteristics and timing, is that deposits H and I were deposited in response to shaking as a result of the 1873 CE Brookings earthquake. This deposit sequence is followed by a flood deposit that is interpreted to have removed subaerial landslide material from the watersheds into the lake after the earthquake.

For this study, we used the distinctive characteristics of deposit J to identify other potential Cascadia earthquake deposits downcore. These characteristics are:

1. Light-coloured (Munsell colour: 2.5Y 4/1, indicating a Slickear Creek watershed source), well-sorted basal silt without visible mica grains and lacking organic matter (such as rootlets).
2. Evidence of loading; projections of the basal silt into the organic-rich sediment below.
3. Presence of a long (at least 2-5 cm but varies with location in the lake) organic-rich tail showing complex grading.

The characteristics of this deposit suggest lower frequency (<5Hz) ground motions capable of destabilizing the subaquatic portion of the lake delta, resulting in a possible small micaceous bypass turbidite that leaves a higher concentration of Slickear Creek watershed silt to be partitioned in the water column during shaking, then finally settling, along with lake organic matter, to form the deposit tail.

We then used the characteristics of deposits H and I, inferred to be the result of a subaerial landslide and subsequent delta failure from the 1873 CE Brookings earthquake, to identify other earthquakes of the same type in the downcore record. The characteristics of deposits H and I used here are:

1. Dark-coloured (Munsell colour: GLEY2 4/5PB) medium silt turbidite present in all cores with visible mica grains.
2. Sharp basal contact without evidence of loading.
3. The schist-derived silt is followed by a light coloured (Munsell colour: 2.5Y 4/1), Slickear Creek watershed-sourced silt (similar to deposit J). This is followed by a flood deposit that is interpreted to reflect the post-seismic removal of landslide material from the watersheds.

The characteristics of this deposit suggest higher frequency (>5Hz) ground motions capable of causing subaerial slope failures resulting in a schist-derived turbidite (indicating a subaerial landslide), followed by light coloured Slickear Creek sourced sediment (indicating a subaquatic delta failure, possibly with an influence from liquefaction). The deposit sequence is followed by a flood deposit (deposit G in Morey et al., 2024, this volume) with mixed composition that likely reflect the removal of post-seismic watershed sediment.

Estimates of ground motions at the site for the 1873 CE Brookings earthquake and the 1700 CE Cascadia earthquake are as follows:

1873 CE Brookings earthquake: Felt reports suggest that this was an earthquake of ~M7. This is equivalent to an MMI of VIII (strong shaking), which reflects a peak ground acceleration of 34-65 %g. Shaking from an earthquake of this magnitude is likely to last up to ~1 minute.

: There are no felt reports for this earthquake of ~M9. The M9 scenario (https://earthquake.usgs.gov/scenarios/eventpage/gllegacycasc9p0expanded_se/executive#shakemap) suggests an M9 earthquake is likely to produce strong ground motions (MMI = VI) at this site, with peak ground accelerations of 9.2-18 %g. Shaking from a full-margin rupture could last as long as 5-6 minutes.

## 3 Results

### 3.1 Identification of earthquake-generated disturbance deposits in Lower Acorn Woman Lake

Disturbance event deposits were identified in the sediment cores used in this study (Table 1) as described in the methods section, then expressed as a diagram (Figure 4). Distinctive beds were correlated using the age data, sedimentology, and physical property data (described further below). Shallow water cores and cores from the northern portion of the lake have less sediment between time equivalent horizons than do the deep water and southern lake cores. The deep-water cores contain thicker disturbance deposits and contain slumps and folds, and occasionally woody debris and portions of soft, partially degraded logs associated with the slumps and folds (labelled on the diagram). A core from the northern site (composite core SQB1/2/ss) was selected to create the chronology to avoid these disturbances in the southern lake cores, avoid an influence from the landslide to the south, and reduce the influence from the second watershed (via Acorn Woman Creek).

Deposits suspected of being triggered by Cascadia earthquakes were identified in the composite core SQB1/2/ss using the characteristics of deposit J as described in the Methods section. Five deposits (deposits K, N, O, R, and X; see Figure 5) were identified as most similar to deposit J in the downcore record. These disturbance deposits are all light-coloured silt layers with few or no visible mica grains, show some evidence of loading into the organic sediment below, and have organic-rich tails exhibiting complex grading. Deposits K and N in the upper portion of the core (Figure 5, bottom left) are most similar to deposit J. These deposits have a well-sorted medium silt unit that bleeds (fine-grained loading) into the organic sediment below. Deposits O, R and X also have similar characteristics, but are slightly different from, deposit J. For example, deposit O is a thinner, light coloured silt with little evidence of loading, deposit R contains evidence of loading as medium silt finger-like projections that are broken off, and deposit X is a large (30 cm long) turbidite showing unusual grading characteristics (including layers of plant macrofossils and benthic diatoms). These details can be seen in Figure 5.

Other possible earthquake deposits identified in Morey et al. 2024 (this volume) are subaerial schist-derived turbidites similar to deposit I which are followed by lighter coloured sediments similar to deposit H (Type 2 deposits). These layers are turbidites that are visible in the cores as dark grey, graded, medium silt layers with visible mica particles, followed by sediment sourced from the Slickear Creek delta. None of the deposits similar to deposit I, however, are followed by a deposit similar to deposit H (which has multiple pulses of Slickear Creek watershed-sourced sediment in it). Many of these dark grey turbidites are followed by flood deposits). The deposits most similar to the sequence represented by deposit H (and the post-seismic flood deposit) are deposits L and S (which do not have a unit similar to deposit I), however deposits M, Q, U and W share some similarities as well (primarily that they are schist turbidites). There are also other schist layers that occur in close proximity

(even coincident) with those deposits most similar to deposit J. For example, the tail of deposit R contains a schist turbidite

and deposit O has a schist turbidite preceding it and within its tail.

Numerous other layers also exist in the cores. These layers have high radiodensity peaks but are thinner than the other deposits that are suspected to be the result of earthquakes. These disturbances are harder to characterize because they are very thin and therefore difficult to sample, however they do not show evidence of loading and are not composed of primarily of light-coloured Slickear Creek watershed-sourced silt.

The downcore sequence of disturbance deposits (A-Z) identified in SQB2 (Figure 5) are also shown in the SQB1/2/ss composite core as shown in Figure 6a. Disturbance event deposits identified as most similar to deposit J are identified in orange and disturbances composed predominantly of schist are identified in grey. Background sediment is identified by blue, and other disturbances are shown in black. Event-free depths of horizons are identified in black numbers to the left of the core diagram whereas the red numbers indicate interevent thicknesses (in centimetres).

The relationship between the lower lake and upper lake cores are shown in Figure 6b. The thickest events in the upper lake core are those identified by Colombaroli et al. (2018) as being outside the distribution of the other silt layers in the record. These layers, identified as disturbances 1-7 in Figure 6b in the USL 2009 core, are correlated to the most prominent disturbances in the lower lake composite core SQB1/2/ss as shown in Figure 6c. This panel also demonstrates how the age data between cores were used to prepare the final age-depth model.

A comparison of Upper and Lower Acorn Woman Lake radiodensity traces (Figure 6c) demonstrates the strong similarities between records. The thickest event deposits in the Upper Acorn Woman Lake record have a similar timing and frequency as the densest disturbance deposits in the Lower Acorn Woman Lake record, and this relationship is just as strong for the low amplitude variability in the Lower Acorn Woman Lake physical property traces. This similarity allowed for a detailed comparison for age data translation as shown in Figure 6c. The oldest portion of the Upper Acorn Woman Lake record is

assumed to be the temporal equivalent of the deposit dated to 1580 +/-20 BP in the Lower Acorn Woman Lake record. The upper and lower lake records can also be seen flattened to one another: SQB1/2/ss was held constant while the Upper Acorn Woman Lake (USL) core data was transformed to line up correlative beds (Figure 6d). The red bars identify the deposit bases that were flattened to the upper lake core.

### 3.2 Within-lake bed correlation and comparison to Upper Acorn Woman Lake

Disturbance event deposits were initially correlated using physical property data, the age model for core SQB2, and individual ages from other cores. This was straightforward for cores where the same horizon was dated in multiple cores, but this was only the case for the horizon ~1200 BP in cores SQB2, SQB14 and SQB5. The majority of the [14]C ages were from core SQB2 (7 samples), one at the base of SQ5, 3 samples from SQB14 and one sample from SQB10. The basal age from SQB14 and the age from SQB10 were too old to be used to date the horizon they were sampled based on correlation of the cores using the

physical property data.

The physical property data reflect the characteristics and amount (relative to organic matter) of clastic sediment in the cores. Whereas radiodensity and magnetic susceptibility in turbidite deposits from marine sediment cores are primary a function of mineralogy and grain size (Goldfinger et al., 2012), these physical properties reflect mineralogy, grain size and % organic matter in the Lower Acorn Woman Lake cores. For example, deposits that have a higher concentration of Slickear Creek watershed-sourced sediment compared to lake margin schist have higher radiodensity and magnetic susceptibility values, a function of the characteristics of the mineralogy. Likewise, the amount of organic matter in the deposit influences the magnetic susceptibility and radiodensity because there is a smaller percentage of clastic particles. This is also evident for the tail of deposit J (Figure 3a) where radiodensity is a sensitive indicator of the inorganic content. Although magnetic susceptibility and density are correlated downcore, there are regions where radiodensity is low and magnetic susceptibility is very high (see, for example, section 7 - at about ~830 cm - in core SQB9). This suggests the presence of highly magnetic minerals, possibly thin tephra layers that are the result of regional volcanic eruptions.

Physical property data, particularly the sub mm-scale radiodensity, can be used to match patterns in lithology downcore. Some of the cores contain a large amount of gas (methane) as pockets within the cores, which causes HU (radiodensity) values to drop suddenly. This is not a problem for the northern composite core SQBss/1/2. Although the noisy data can complicate matching the downcore patterns, the patterns dominate the radiodensity variability, and it is possible to correlate beds and even low-amplitude signals throughout the lake.

Colombaroli et al. (2018) identify seven thick silt deposits that are outside the frequency magnitude (power law) relationship of silt events in the Upper Acorn Woman Lake core (Colombaroli et al., 2010; 2018). Although these thick units appear homogenous because of similar particle size (medium silt) throughout the deposit (until the very top, where it becomes a silty-clay cap), there are compositional differences that result in a slight increase in radiodensity from the base of the deposit to the top (Figure 9). The very thin (mm-scale) silty-clay cap at the top would be expected if most of the fine sediment suspended in the water column is transferred into Lower Acorn Woman Lake via Acorn Woman Creek. The Upper Acorn Woman Lake deposits do not have an organic tail but are followed by a sequence of progressively thinner silt layers before returning to normal sedimentation. These seven silt deposits can be identified as correlative units in the Lower Acorn Woman Lake record (numbered events in Figure 6).

### 3.3 Lower Acorn Woman Lake age-depth model

The radiocarbon determinations used in the age-depth model are listed in Table 2. Radiocarbon samples in bold were used in the age-depth model, and those in grey were not (determined based on the relationship between sample locations cores and relative ages). The radiocarbon datums were translated from Upper Acorn Woman Lake onto composite core SQBss/1/2 after the stratigraphic sequences were aligned (Figure 6c, d). The age and depth data were then used to create the Lower Acorn Woman Lake age-depth model shown in Figure 7 (using OxCal as described in the methods section using the assumptions presented in Morey et al., 2024, this volume).

### 3.4 Cascadia earthquake chronology

Based on the age-depth model for the Lower Acorn Woman Lake composite core SQBss/1/2, the ages for the deposits identified as likely the result of a major disturbance event are shown in Table 3. Disturbance event deposits S and W are italicized because the deposits have some, but not all, of the characteristics of deposit J. Both deposits show evidence of loading, but do not have a distinctive light coloured silt layer like the other deposits. Deposits S through X have large age ranges because of the limited age control data at those depths in the cores.

## 4 Discussion

### 4.1 Origin of the disturbance event beds

Increases in physical property data (magnetic susceptibility and radiodensity) in lake sediment cores can be the result of a variety of events. These include 1) aseismic events that require water such as post-fire erosion, storm remobilization of sediment, land-use changes and flooding, 2) seismic events which trigger landslides and submarine lake floor sediments, 3) mixed seismic and aseismic events, such as the destabilization and subsequent transport via flooding of destabilized hillslope sediment into the lake, 4) concentrations of authigenic minerals such as magnetite, 5) shaking from nearby volcanic activity, and 6) layers of volcanic tephra.

#### 4.1.1 Post-fire and flood-related erosional events

Lakes throughout Cascadia have been used extensively to reconstruct fire histories and post-fire erosion, where minerogenic layers and charcoal abundance data are frequently used to infer increased erosion after large wildfires (e.g., Millspaugh and Whitlock, 1995; Colombaroli and Gavin, 2010). Although charcoal analysis has been successfully used as a proxy for post-fire erosional events, Long et al. (1998) show that peaks in charcoal accumulation do not always correlate with magnetic susceptibility at Little Lake, Oregon. Similarly, charcoal peaks at Bolan Lake, Oregon, and Sanger Lake, California, (between the coast and Acorn Woman Lakes at a similar latitude) are not always associated with (or immediately precede) magnetic susceptibility peaks (C. Briles, pers. comm., 2010). Although wildfires are common in Cascadia and charcoal analysis at nearby Upper Acorn Woman Lake suggests that wildfires do influence erosion in this region (Colombaroli and Gavin, 2010), subsequent analysis of the pseudo-annual silt layers observed in the radiodensity data has suggested that the seven largest disturbance events in the record are the result of a different process at Upper Acorn Woman Lake other than erosion, possibly earthquakes (Colombaroli et al., 2018). Both floods and post-fire erosion, however, may be a source for the smaller deposits in the Lower Acorn Woman Lake record other than those that are correlated to the seven thickest deposits in the Upper Acorn Woman Lake record. Flood disturbances have been identified in the Lower Acorn Woman Lake record (presented and discussed in Morey et al., 2024, this volume) as deposits

with slight increasing, then decreasing, grain size and % clastics with a physical property profile that is somewhat rounded in appearance, typical of the waxing and waning of storm events (Mulder, 2001).

### 4.1.2 Volcanism

Three volcanoes are within 100-130 km of Acorn Woman Lakes: Mt. Shasta and Medicine Lake in California, and Crater Lake in Oregon. Eruptions from these volcanoes are known to produce tephra layers in sediments in lakes to the east (because prevailing winds are generally from the west) but are unlikely to produce significant layers in Acorn Woman Lakes because they are to the west of the volcanoes. Cryptotephra layers however, although not visible by eye, are likely to leave evidence behind in lake sediments. These layers can have high magnetic susceptibility values even though sediment density is low (Boes

et al., 2017). An example of this is at ~830 cm in core SQB9. Although not confirmed to be cryptotephra, these layers are not likely to be confused with earthquake triggered deposits because they are not dense, are not visible silt layers with tails, yet have high magnetic susceptibility.

Shaking from volcanic activity could also influence lake sediments, however the impact would likely be minor at Acorn Woman Lakes because of the distance from the volcanoes (100 km or more), and because ground motions are likely to be

lower than the threshold (~MMI VI) required to cause lake disturbances at these distances.

### 4.1.3 Landslides

Landslides unrelated to earthquakes can also leave evidence in lakes. Deposit E suggests that landslide deposits can be identified by their lack of areal extent, however larger landslides created the lakes when it blocked Acorn Woman Creek near its confluence producing lake-wide effects in both lakes. The landslide appears to be retrogressive, forming Lower Acorn

Woman Lake just prior to deposit Z (which is dated to 2580 (2370-2700) BP), and Upper Acorn Woman Lake at 1490 (1410-1530) BP. These types of landslides also are unlikely to be confused with earthquake deposits (although earthquakes may have triggered the landslides) because they resulted in lake formation and are not layers in an existing lake. Submerged trees exist in each lake and could be sampled and analysed to determine the exact time each lake was formed.

### 4.1.4 Earthquakes

This study seeks to differentiate between plate boundary earthquake from non-plate boundary earthquakes and aseismic disturbance deposits by comparing the record of disturbances from Acorn Woman Lakes to published records of Cascadia megathrust earthquakes. To evaluate the relationship between disturbance deposits similar to deposit J and the marine record, we first use what is known from the historic portion of the record to suggest interpretations based on the sedimentology (Morey et al., 2024, this volume). We then compare the temporal relationship between the Lower Acorn Woman Lake record and

nearby paleoseismic records of Cascadia earthquakes to look for evidence of synchronous triggering. Our initial observation is that the timing and frequency of disturbance deposits composed of Slickear Creek watershed-sourced silt (shown in Figure 6 and listed in Table 3) are similar to the timing and frequency of Cascadia earthquakes. The thickest of the marine beds

correlate to the thickest beds from the Upper Acorn Woman Lake record. If bed thickness is a shaking duration proxy, then they might be expected to correlate.

### 4.2 Deposits similar to Deposit J

#### 4.2.1 Temporal relationship to regional paleoseismic records

Figure 6a shows the 26 disturbance deposits, DE's A-Z, with large excursions in magnetic susceptibility and radiodensity in the downcore record from Lower Acorn Woman Lake. Eight of these disturbance event deposits (Table 3; including deposit J) have some of the characteristics of deposit J, which was suggested in Morey et al., 2024 (this volume), to have formed in response to ground motions from the 1700 CE Cascadia earthquake. The other disturbance deposits identified in the downcore record are also of higher radiodensity and magnetic susceptibility compared to background sediment, but do not have a distinctive Slickear Creek watershed sourced composition or some of the other characteristics of deposit J.

Table 4 identifies the beds from Lower Acorn Woman Lake and possible correlatives from marine and coastal paleoseismic data – the T numbers in bold are the thickest beds at Rogue Apron site which are also those deposits that correlate to beds at Hydrate Ridge West (Figure 8; both of which are marine turbidite sites from Goldfinger et al., 2012). This suggests that the disturbances with Slickear Creek watershed-sourced silt, organic tails, and evidence of loading from the Lower Acorn Woman Lake record are most likely the result of significant plate boundary earthquakes (as described in Goldfinger et al., 2012).

Table 4 indicates that there is an excellent temporal match between the four coastal, lake and marine paleoseismic sites for the margin-wide events T1, T2, T3, T4, and T5 (using the marine turbidite bed notation). T1/deposit J, T3/deposit N, and T4/deposit O correlatives have radiocarbon determinations which are within a few decades, whereas there is poorer agreement between T2/deposit K and T5/deposit R medians and ranges even though the ranges overlap significantly (some ranges are simply larger than others). Although DE-X appears to correlate to T6 (based on a comparison of physical property data between lake and marine cores), the age range for DE-X is large compared to the range for the marine sites (460 yrs for deposit X compared to a few hundred years for Rogue and Hydrate Ridge sites) making this linkage less certain. A distinctive ~1000 yr gap (Atwater 2004; Kelsey et al., 2005; Goldfinger et al., 2012; and Witter et al., 2012) occurs in the record of Cascadia earthquakes between T5 and T6 in all records, and this can be seen in the Lower Acorn Woman Lake record as well. A graphical representation of the relationships between these age distributions can be seen in Figure 8.

#### 4.2.2 Correlation of physical property data between Lower Acorn Woman Lake and the marine paleoseismic record

The Lower Acorn Woman Lake record was correlated to marine sites Rogue Apron and Hydrate Ridge Basin West (Goldfinger et al., 2012; see Figure 1 for core locations) using physical property and radiocarbon data. The relationships between cores are shown in the bed-flattened correlation diagram (Figure 9). This method of correlation appears to work even though the physical property data for the marine cores typically reflects the amount of magnetic minerals and grain size of horizons downcore,

whereas the physical property data is also influenced by the percentage of organic matter (which can be part of the graded sequences) and clastic characteristics in the lake core.

### 4.2.3 Changes in deposit characteristics downcore

The characteristics of deposits similar to deposit J, in particular deposits N and X, show evidence of partitioning during shaking as a result of earthquake ground motions. Deposits N and X are partitioned from the base upward from a base of Slickear Creek watershed-sourced silt to a tail containing a debrite containing organic matter (in the case of deposit X), then a layer of benthic diatoms (all of similar size; observed in both deposits N and X). These features are likely not the result of simple differential settling because they are not observed in other types of turbidite tails but are more likely to be the result of partitioning in the water column as a result of sustained ground motions. This suggests that the presence of complex grading is a distinctive characteristic of subduction earthquake deposits at this location.

The presence of load structures downcore do not appear to be a distinctive characteristic of subduction earthquake deposits. The deposit sequence that includes deposits S1 (interpreted to be a subduction earthquake correlating to T5a), S2 (immediately below deposit S1, interpreted to be a crustal earthquake deposit), and the deposit below S2 (possibly another crustal earthquake deposit) display load structures. It appears likely that shaking from any earthquake type may produce these load structures, depending on the size fraction of the overlying silt, and therefore these are not simply load structures.

### 4.3 Deposits similar to deposits H and I and smaller disturbance event beds

There are smaller deposits in the Acorn Woman Lake sequence that were not identified as potential Cascadia megathrust earthquake deposits because they do not have the characteristics of deposit J. Some of these correlate to the large number of smaller deposits in the marine record. Deposit L is one example. This deposit is a dark grey turbidite with visible mica flakes, and therefore is more similar to the wall failure deposit E and seismogenic turbidite deposit I described in Morey et al., 2024 (this volume). It is followed by a flood deposit that suggests the post-seismic removal of watershed sediment, but does not have the equivalent of deposit H. This deposit has a correlative unit in the Upper Acorn Woman Lake record (dated to 550-670 BP, two-sigma range, Figure 6) and is contemporaneous with marine event T2a. The different composition (schist-derived) and characteristics (it is a lake-wide turbidite) of this deposit in the lower lake core suggests that it was not a Cascadia earthquake that triggered this deposit, but likely a different type of earthquake deposit. It also does not match the timing of earthquakes on the northern San Andreas fault (nor do T2b, T3a, or T4a; see Goldfinger et al., 2019, 2020). For these smaller events our uncertainty is higher because we can't use size of the event (because frequency is a function of distance) for correlation, and there is more uncertainty in timing. These events, then, are harder to interpret. Based on the available information, however, our preferred interpretation is that the thickest of these subaerially sourced schist-derived deposits, including deposit L, are the result of earthquakes with subaerial landslides and not complex deposits in response to shaking from a Cascadia earthquake. Without further information, because T2a, T2b, and T4a are not San Andreas fault earthquakes and they are present in the Upper and Lower Acorn Woman Lakes records as well as at the marine Rogue site, it is suggested

that they may be the result of activity on regional crustal fault earthquakes. Because the Canyonville fault (see Figure 1 in Morey et al., 2024, this volume) extends from near Acorn Woman Lakes to the head of the Rogue channel, it seems plausible that activity on this fault explains the presence of these and other smaller deposits at both sites.

The downcore record does not include any deposit sequences exactly like deposits H and I, and in fact deposits like H and I are decoupled downcore. Deposit R is similar, but reverse from deposits H and I, with a light-coloured silt deposit first and the

crustal earthquake (schist-derived) deposit within the deposit tail (suggesting very close – likely hours to days – timing). The ones most similar have characteristics of deposit I and the post-seismic flood deposit, but do not have an associated Slickear Creek watershed-sourced deposit similar to deposit H. This suggests that the 1873 CE Brookings earthquake was different from any other type of earthquake represented in the past or that conditions were unique at that time. Given that it is most likely that conditions at the lake have likely been similar through time, it is considered more likely that the earthquake was

unique. No other deposit shows evidence of multiple ruptures, and none of them have a massive silt unit sourced from the Slickear Creek watershed. The similar deposits suggest a wall failure deposit followed by a delta failure (which is a mixture). Because deposit H is somewhat similar to other downcore deposits similar to deposit J, this seems to confirm that the 1873 CE Brookings earthquake had a subduction component.

Other sources of seismicity may also influence these records: earthquakes on other crustal faults (such as the San Andreas fault

or other unidentified regional faults), the Gorda Plate, nearby transform faults, and intraplate earthquakes. There have been suggestions that both the M7.9 1906 CE San Andreas earthquake and the ~M7.0 1873 CE Brookings intraplate earthquake are found in both the marine and lake records (Morey et al., 2024, this volume, and Goldfinger, 2019; 2020), and similar events are likely the source of some of the smaller beds. Further exploration of these small event beds is beyond the scope of this paper, however the presence of active seismicity on regional crustal faults has significant hazard implications for the region.

**4.4 Data integration**

**4.4.1 Regional data relationships**

The correlation between the Lower Acorn Woman Lake disturbance record and Rogue Canyon and Hydrate Ridge marine sediment cores (Figure 9) suggests that all the full-margin ruptures (T1-T6), including T2 (using the marine turbidite T numbers from Goldfinger et al., 2012), younger than 2700 BP disturb sediments in Lower Acorn Woman Lake. T2 is unusual because

it is not found in any of the coastal southern Cascadia paleoseismic sites (Coquille River, Bradley Lake, and Sixes River; Kelsey et al., 2005; Witter et al., 2012) but is found in the marine turbidite record of Goldfinger et al. (2012; 2013) and at one site in Padgett et al. (2021). These southern coastal tsunami sites require that the tsunami must have been large enough to have entered the lake or estuary. Megathrust earthquakes that produced turbidite deposits T1, T3, T4, T5 and T6 caused large tsunamis at Bradley Lake (Figure 8). It is possible that T2, a smaller turbidite, may have been triggered by a smaller earthquake

which produced a smaller tsunami that did not overcome the threshold to leave a deposit at Bradley Lake or other coastal sites.

Although T3a, a southern Cascadia event, was not identified as an individual disturbance event deposit in the Lower Acorn Woman Lake record, there is a disturbance event with a smaller magnetic susceptibility and radiodensity signature in the core, suggesting a small event deposit. This region is complex tectonically, and further investigation is needed to determine if this and similar deposits are the result of crustal earthquakes, small southern Cascadia earthquakes, or intraplate earthquakes.

### 4.4.2 Timing of lake formation

T5 and T6 were deposited at similar times as Lower Acorn Woman Lake (below deposit Z dated 2370-2700 BP; ~T6) and Upper Acorn Woman Lake (1410-1530 BP; ~T5) were formed. Did shaking from a megathrust earthquake cause the landslide to fail, creating the lakes or were they triggered by ruptures on nearby crustal faults? High-frequency ground motion from a nearby crustal earthquake is suspected in Morey et al., 2024 (this volume), to be the cause of the subaerial lake-wide deposit attributed to the 1873 CE Brookings earthquake, suggesting that local earthquakes can cause landslides at this location, however it is also possible that the landslide is unrelated to earthquakes. This is important because there have been several studies that have attempted to link landslides in the Coast Range and elsewhere to Cascadia earthquakes without success (see, for example, Struble et al., 2020 and LaHusen et al., 2020).

### 4.4.3 Lake and deposit characteristics

Based on this study the following characteristics of lakes are suggested as optimal sites for paleoseismic studies in Cascadia:

1.  Small lakes ($< 1$ km$^2$), to reduce the influence of localized disturbances more common in large lakes.
2.  Landslide dammed lakes, to ensure sufficient sedimentation rates.
3.  High sedimentation rates (~1-2 cm/decade); mixed clastic and organic (roughly 50% each) sedimentation.
4.  The presence of a delta.
5.  Water depth greater than ~7 meters to prevent an influence from bioturbation.

Based on this study, the revised characteristics of Type 1 (subduction earthquake) and Type 2 (crustal earthquake) deposits found useful are as follows:

Type 1 deposits:

1.  Sediment remobilized from the delta slope, with a component that is enriched in Slickear Creek watershed-sourced sediment.
2.  Complex grading, including a long (but highly variable) organic tail.
3.  No evidence of subaerial landslides (unless the Type 1 and Type 2 deposits are coeval or nearly so).

Type 2 deposits:

1.  Subaerially sourced turbidite containing schist-derived sediment.
2.  The resulting turbidite is followed by post-seismic removal of sediment as a result of flooding.

This reinterpretation of Type 2 deposits suggests that the pulses present in deposit H are unlikely to be the result of a crustal earthquake sequence because it is composed of Slickear Creek watershed-sourced silt (more than any other deposit in the

historical portion of the record) and not derived from the schist bedrock surrounding the lake. This suggests that the pulses may actually be related to ground motion variability as a result of a southern Cascadia subduction earthquake, producing a tsunami that was not large enough to enter Bradley Lake.

## 5 Summary and Conclusions

Six to eight of the 26 disturbance event deposits identified in this chapter have the characteristics of deposit J, which has been attributed to the 1700 CE Cascadia megathrust earthquake. These disturbances correlate to the thickest disturbances in the Upper Acorn Woman Lake record, and to full margin megathrust events onshore coastal and offshore marine turbidite records over the past 2700 years: T1, T2, T3, T4, T5 and possibly T6 (as interpreted by Goldfinger et al., 2012), and possibly T5a and T5b. There are smaller events in the Lower Acorn Woman Lake record as well. T2 and T2a marine equivalents are missing in the tsunami record from Bradley Lake, however both have temporal equivalents present in Lower Acorn Woman Lake and may be the result of intraplate or crustal earthquakes. Many of these smaller deposits cannot be attributed to specific events, however there is the potential to sort this out in the future with further analysis and additional radiocarbon ages. For example, although T3a was not identified as a significant disturbance in the Lower Acorn Woman Lake record, there is a smaller disturbance below T3 that could be dated to determine if it is contemporaneous with T3a in the offshore record.

Some of the smaller events are schist-derived turbidites which appear to be independent responses to a single event not part of a complex sequence as interpreted in Morey et al., 2024 (this volume). These may be the result of ruptures on regional crustal faults. Table 4 shows that the Lower Acorn Woman Lake record has the temporal equivalents of T2a, T4a, T5a, T5b and T5c. The only event missing in this time range is T3a. Of these, T2a, T4a and T5b are interpreted as possible crustal events, while T5a and T5c equivalents are more similar to plate boundary earthquakes. So, it seems possible that there are mixed sources for these smaller southern events, but it is not possible to tease them apart without more information.

These results lead us to come to three primary conclusions.

1) The timing and frequency of the largest event deposits composed of light-coloured Slickear Creek watershed-sourced silt from Lower Acorn Woman Lake to the offshore and coastal records of Cascadia earthquakes is strong evidence that Lower Acorn Woman Lake is a good recorder of Cascadia earthquakes.

2) The interpretation that southern Cascadia earthquakes are more frequent may be in part a result of crustal fault earthquakes, such as the influence of crustal faults that cut the head of channel systems. This suggests that most of the Cascadia earthquakes over the past 2,700 years are full-margin ruptures.

3) Evidence of crustal earthquakes suggests a previously unknown hazard from crustal faults in southern Cascadia.

**Acknowledgements**

This research was partially supported by the National Earthquake Hazards Reduction Program of the U.S. Geological Survey (through a grant to Andrew Meigs and Simon Engelhart, USGS Grant G17AP00028). The USGS Earthquake Hazards Program through Alan R. Nelson (USGS, Golden) partially supported the collection of some Livingstone cores and CT scans of Livingstone and Kullenberg cores. Geological Society of America awards provided additional funding: a graduate student research grant and the Kerry Kelts Limnogeology Award. Coring in 2015 would not have been possible without contributions from Joseph Stoner, Roy Haggerty, and a donation from Ruth Morey.

The US Forest Service granted permission for this study (special thanks go to Starr Ranger Station employees and the regional office in Grants Pass, OR). We are extremely grateful for the assistance and knowledge provided by Ranger John McKelligott, and for field assistance by Mark Anthony (USFS employee who participated in coring in 2015). Bert Harr generously allowed access to his Slickear Creek property during this investigation and contributed significantly to this project through his vast knowledge of local extreme events and local and regional history since 1900. Peter Jones provided personal historic accounts of the more recent historic floods.

This project could not have happened without the generous assistance from numerous volunteers. Dan Gavin (UO) and Alan R. Nelson (USGS-Golden) provided coring equipment, expertise, and guidance during this project. Katie Alexander (Western Washington University) spent a few days canoeing in the cold to acquire bathymetric data. Maureen Walczak (OSU) generously analyzed my first radiocarbon samples and provided guidance on how to use the radiocarbon production curve to select samples. Jamie Howarth (then at GNS, NZ) provided useful coring information and guidance, including sharing his approach to dating an earthquake event in lake sediments from about the same time as the 1700 CE Cascadia earthquake. Other volunteer field assistants included Randy Keller, Brendan Reilly, Katie Alexander, and many others. Christy Briles (Colorado University, Denver) helped train me in the fine art of lake coring during a fateful summer week in 2010.

LacCore and the University of Minnesota, provided Kullenberg coring equipment and expertise. Mark Shapley, my LacCore contact, provided guidance, knowledge, and enthusiastic discussions about data. Thanks also go to the OSU core repository (especially Maziet Cheseby) for housing cores and providing the tools to process them. Carol Chin aided core processing of the Kullenberg cores, for which I am extremely grateful.

*Author contributions:* AEM conceptualized the idea, developed the methodology, conducted the analysis, and wrote the original paper, and CG contributed by providing offshore turbidite data, gave advice when comparing results to other paleoseismic sites, and provided suggestions as to presentation of results. Both authors reviewed and edited the paper for the final version.

*Competing interests:* The contact author declares that neither author has competing interests.

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

**Table 1.** *Sediment Cores.* USL = Upper Acorn Woman Lake; SQB = Lower Acorn Woman Lake. Sediment core locations, water depth, and sediment-core lengths are listed for all cores used in this study. Cores highlighted in bold text are the primary core sites. Cores SQB6 and SQB7 are missing the historic portion of the record.

| Core name | Type | Length (m) | Water depth (m) | Latitude (°) | Longitude (°) |
|---|---|---|---|---|---|
| SQB-ss | Surface core | 0.80 | 16.9 | 42.04405 | -123.01853 |
| **SQB1** | **Livingstone** | **6.74** | **16.9** | **42.04405** | **-123.01853** |
| **SQB2** | **Livingstone** | **7.37** | **16.5** | **42.04405** | **-123.01853** |
| **SQB5** | **Livingstone** | **3.98** | **23.5** | **42.04264** | **-123.01909** |
| SQB8 | Kullenberg/Gravity | 8.01 | 30.0 | 42.04227 | -123.01908 |
| SQB9 | Kullenberg/Gravity | 8.29 | 37.0 | 42.03982 | -123.02050 |
| SQB10 | Kullenberg/Gravity | 10.08 | 35.0 | 42.03857 | -123.02108 |
| SQB11 | Kullenberg/Gravity | 7.55 | 29.2 | 42.03778 | -123.02175 |
| SQB12 | Kullenberg/Gravity | 5.24 | ~20.0 | 42.04191 | -123.01864 |
| SQB13 | Kullenberg/Gravity | 6.24 | 25.0 | 42.02056 | -123.02056 |
| **SQB14** | **Kullenberg/Gravity** | **8.28** | **30.0** | **42.04356** | **-123.01836** |
| SQB15 | Kullenberg/Gravity | 4.55 | 28.5 | 42.04197 | -123.01945 |
| USL | **Livingstone** | 10.2 | 14.1 | 42.19167 | -123.09333 |

**Table 2.** *Radiocarbon ages in radiocarbon years, and the [137]Cs peak (for Upper Acorn Woman Lake core only; Colombaroli et al., 2010; 2018).* The samples in grey text (Samples 0, and 11-13) were not included in the age model because they are inferred to be reworked. Samples in bold text were used to create the age-depth model for the historic portion of the sequence. Upper Acorn Woman Lake depths are composite depths from splicing together two cores (Upper Acorn Woman Lake I and Upper Acorn Woman Lake II; collected in 2009) with overlapping 1-m drives to produce a continuous sequence; Colombaroli and Gavin, 2010. NOSAMS = National Ocean Sciences Accelerator Mass Spectrometry, SQB = Lower Acorn Woman Lake, USL = Upper Acorn Woman Lake. Sample ID's include the original sections and depths (archival and event-free composite) for the SQB cores, and composite depth for the USL cores.

| Sample # | ID | Description | Laboratory and sample no. | [14]C yrs BP |
|---|---|---|---|---|
| 0 | SQB1A; 14.0-14.5 cm | Fir needle | S-ANU 42418 | 865+/-35 |
| **1** | **SQB1A; 15.5-16.0 cm event-free: 65 cm** | **Fir cone frag** | **S-ANU 42419** | **255+/-25** |
| **2** | **SQB1A; 25.5-26.0 cm Event-free: 71 cm** | **Fir needle** | **S-ANU 42618** | **110+/-25** |
| **3** | **SQB1A; 35.5-36.0 cm Event-free: 81 cm** | **Fir needle** | **S-ANU 42617** | **190+/-25** |
| **4** | **SQB1A; 84.0-85.0 cm Event-free: 108 cm** | **Fir needle** | **S-ANU 42616** | **260+/-40** |
| **5** | **SQB1A; 95.0-96.0 cm Event-free: 115 cm** | **decid. Plant frags** | **S-ANU 42417** | **630+/-25** |
| **6** | **SQB1B; 67.0-68.0 cm Event-free: 185 cm** | **plant frags** | **UCIAMS 140214** | **1155+/-20** |
| **7** | **SQB2H; 39.0 cm Event-free: 566 cm** | **plant frags** | **OS-109825** | **2480+/-20** |
| 8 | SQB5C; 27-28 cm | Cone bract | NOSAMS | 1270+/-20 |
| **9** | **SQB5D; 99-100 cm Event-free: 336 cm** | **Cone bract** | **NOSAMS** | **1580+/-20** |
| 10 | SQB14 sec 2, 81cm | Fir needle | NOSAMS | 1220+/-20 |
| 11 | SQB14 sec 3; 122.5cm | Twig | NOSAMS | 2310+/-20 |
| 12 | SQB14 sec 6; 30.5-31cm | Deciduous leaf | NOSAMS | 4470+/-25 |
| 13 | SQB10 sec 3, 54-55cm | Plant fragment | NOSAMS | 1810+/-20 |
| **14** | **USL; [137]Cs** | **Bulk samples** | **Flett Research, Inc.** | **~1964 CE** |
| 15 | USL; 539.5 cm | Charred wood | NOSAMS 64498 | 615+/-40 |
| **16** | **USL; 630.5 cm Event-free: 145 cm** | **Terrestrial plant macros** | **NOSAMS 64497** | **980+/-55** |
| 17 | USL; 729 cm | Wood | Beta-23617 | 1110+/-40 |
| 18 | USL; 856.5 cm | Bud scale | NOSAMS 64496 | 1610+/-140 |
| 19 | USL; 952.5 cm | Douglas-fir needle | NOSAMS 64495 | 1870+/-100 |

**Table 3.** *Calendar ages for deposits with characteristics of deposit J. Given are calibrated mean, mode and age ranges (95%* confidence) in BP for the disturbance event deposits suspected to be Cascadia earthquakes in composite core SQBss/1/2. Those disturbance deposits in italics are less similar to deposit J as compared to the other deposits listed.

| Event ID | depth (event-free; cm) | Cal yr BP Median | Cal yr BP Mean | 2-sigma age range (Cal yr BP) |
|---|---|---|---|---|
| DE-H | 58 | 100 | 100 | 60-130 |
| DE-J | 71 | 230 | 230 | 170-270 |
| DE-K | 101 | 370 | 360 | 280-460 |
| DE-N | 148 | 860 | 870 | 750-980 |
| DE-O | 207 | 1190 | 1200 | 1090-1280 |
| DE-R | 336 | 1490 | 1490 | 1430-1530 |
| *DE-S* | *383* | *1710* | *1700* | *1520-1920* |
| *DE-W* | *496* | *2250* | *2250* | *2010-2500* |
| DE-X | 522 | 2370 | 2380 | 2150-2600 |

**Table 4.** *Comparison of Lower Acorn Woman Lake chronology to those from Hydrate Ridge, Rogue and Bradley Lake records.* Bold event IDs at left indicate those with characteristics similar to deposit J, whereas bold T#'s indicate thickest Rogue Apron beds. Blue ages are hemipelagic-derived ages whereas black indicates ages based on radiocarbon (but corrected to reflect the age of the deposit). For more information on marine ages see Goldfinger et al., 2012 (local ages, not averages, were used). The Bradley Lake data are the modified versions from Goldfinger et al., 2012. This was to use the youngest, rather than average, ages.

| Event ID | Lower Acorn Woman L. | Event | Bradley Lake | T# | Rogue Apron | Hydrate Ridge |
|---|---|---|---|---|---|---|
| **DE-H** | 100 (60-130) | | | | | |
| **DE-J** | 230 (170-270) | DE-1 | 250 | **T1** | 250 (200-300) | 300 (230-410) |
| **DE-K** | 370 (260-460) | | | T2 | 490 (380-590) | 509 (410-610) |
| DE-L | 590 (550-660) | | | T2a | 550 (430-670) | |
| DE-M* | 840 (740-960) | | | | | |
| **DE-N** | 860 (750-980) | DE-2 | 940 (800-1060) | **T3** | 740 (670-810) | 800 (700-910) |
| | | DE-3 | 1010 (930-1090) | T3a | 1070 (970-1200) | 1060 (950-1180) |
| **DE-O** | 1200 (1080-1290) | DE-4 | 1360 (1300-1420) | **T4** | 1200 (1100-1290) | 1210 (1100-1340) |
| DE-P* | 1370 (1260-1470) | | | | | |
| DE-Q | 1390 (1290-1490) | | | T4a | 1370 (1270-1500) | 1470 (1330-1620) |
| **DE-R** | 1490 (1410-1530) | DE-5 | 1520 (1370-1630) | **T5** | 1560 (1400-1730) | 1650 (1490-1813) |
| *DE-S-1** | 1700 (*1520-1930*) | DE-6 | 1540 | T5a | 1760 (1580-1930) | |
| *DE-S-2** | undated | | | | | |
| DE-T | 2020 (1760-2290) | | | | | |
| DE-U | 2120 (1860-2390) | | | T5b | 2020 (1850-2180) | |
| DE-V | 2180 (1910-2430) | | | | | |
| *DE-W* | 2250 (*2000-2500*) | | | T5c | 2320 (2180-2470) | |
| **DE-X** | 2380 (2140-2600) | DE-7 | 2550 (2350-2618) | T6 | 2560 (2490-2710) | 2450 (2410-2720) |
| DE-Y | 2510 (2340-2700) | | | | | |
| DE-Z | 2580 (2370-2700) | | | | | |
| | | | | T6a | 2730 (2590-2880) | |
| | | | | T6b | 2820 (2680-2990) | |
| | | DE-8 | 3120 (2960-3260) | **T7** | 3060 (2860-3220) | 2960 (2820-3070) |

*Lower Acorn Woman Lake DE's identified in grey (DE-M and DE-P) may be linked to DE's N and Q and not separate events. DE-S is separated into two (possibly three) individual events. The upper event is dated because it contains Slickear Creek watershed-sourced sediment, whereas the lower ones do not. DE's in italics have inconclusive sedimentological characteristics as compared to deposit J.

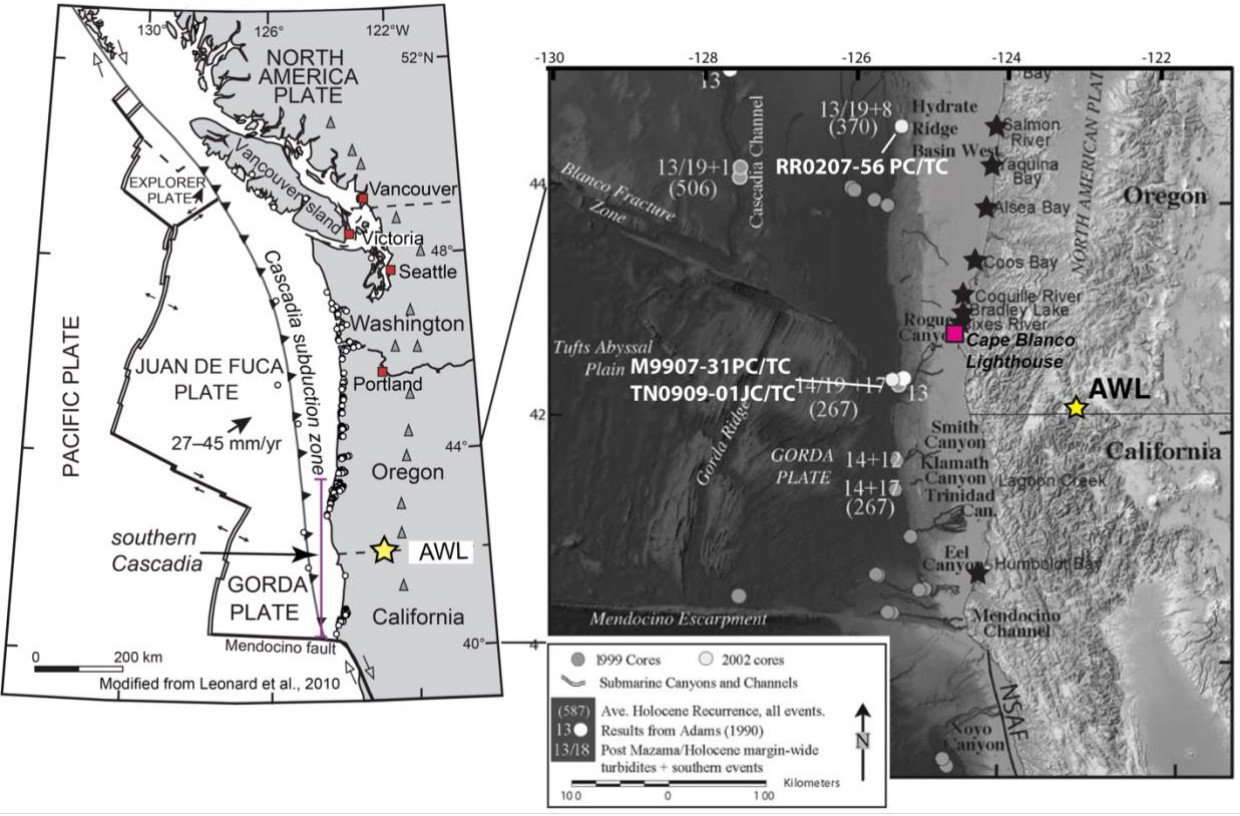

Figure 1. *Location map*. The yellow star identifies the location of Acorn Woman Lakes (AWL) used in this study. Black stars
indicate the coastal paleoseismic sites presented in this study. The base map (right) identifies the location of channel systems
and sediment cores used to reconstruct the offshore record of Cascadia earthquakes (Goldfinger et al., 2012).

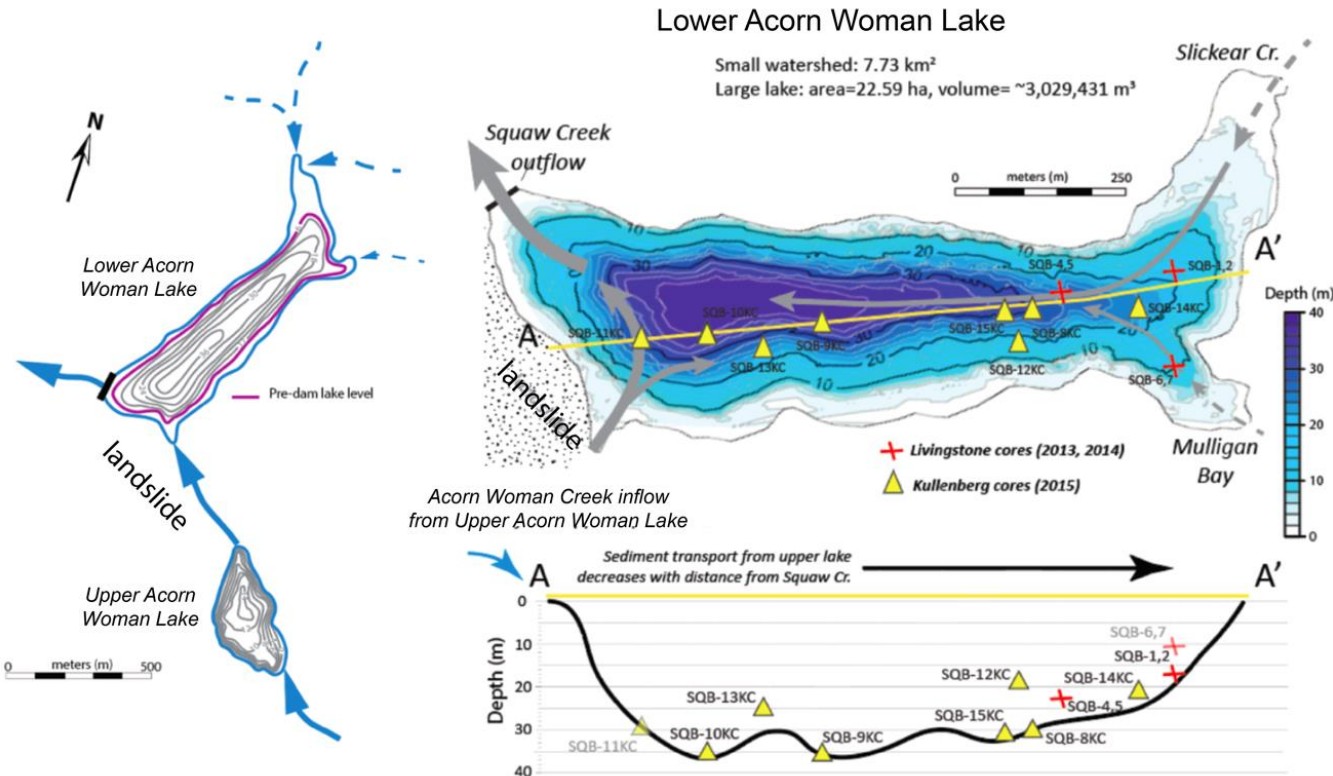

Figure 2. *Lake Setting*. Upper and Lower Lakes are shown with core locations. Left: The lakes are connected hydrologically by a small stream (Acorn Woman Creek) that crosses a portion of the landslide that created Upper Acorn Woman Lake. Right: The core locations for each of the sediment cores from Lower Acorn Woman Lake are identified by yellow triangles (2015 cores) or red X's (2013 and 2014 cores). The Upper Acorn Woman Lake core was taken from 14.1 m water depth, at the lake's depocenter.

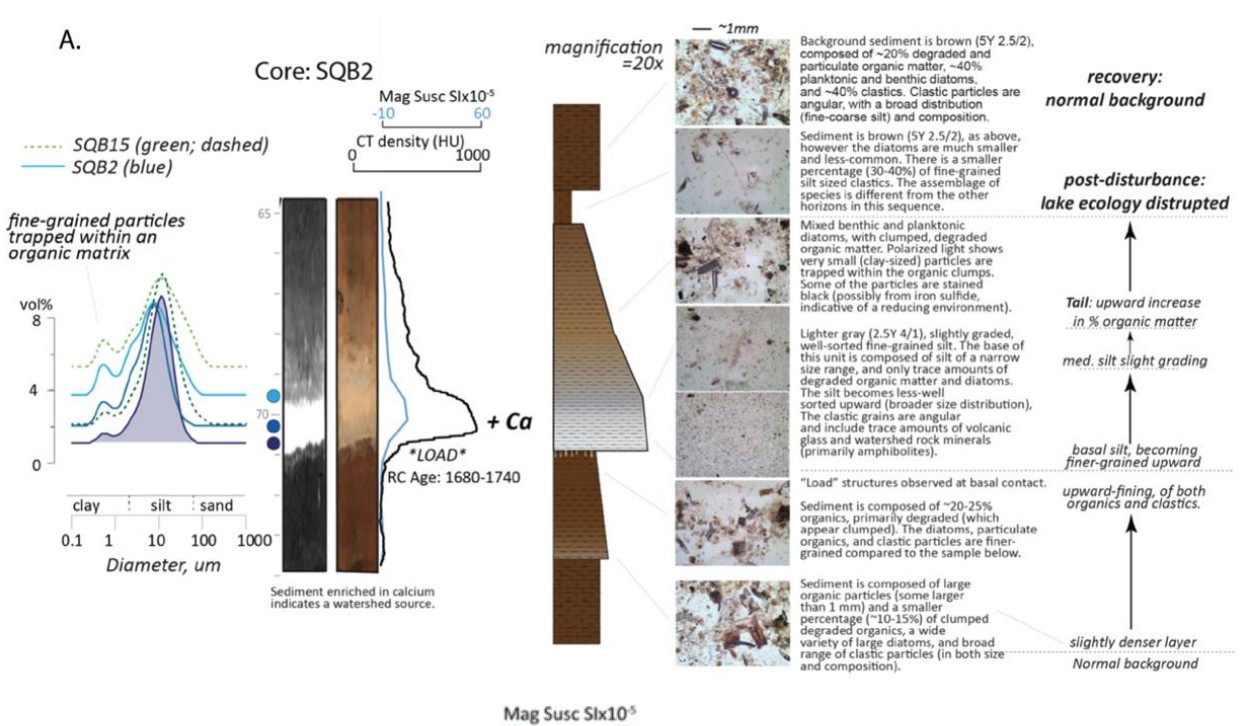

**A.**

Core: SQB2

SQB15 (green; dashed)
SQB2 (blue)

fine-grained particles trapped within an organic matrix

vol%

clay | silt | sand

Diameter, um

Mag Susc SIx10⁻⁵

CT density (HU)

Sediment enriched in calcium indicates a watershed source.

*LOAD*
RC Age: 1680-1740

+ Ca

magnification =20x

— ~1mm

Background sediment is brown (5Y 2.5/2), composed of ~20% degraded and particulate organic matter, ~40% planktonic and benthic diatoms, and ~40% clastics. Clastic particles are angular, with a broad distribution (fine-coarse silt) and composition.

Sediment is brown (5Y 2.5/2), as above, however the diatoms are much smaller and less-common. There is a smaller percentage (30-40%) of fine-grained silt sized clastics. The assemblage of species is different from the other horizons in this sequence.

Mixed benthic and planktonic diatoms, with clumped, degraded organic matter. Polarized light shows very small (clay-sized) particles are trapped within the organic clumps. Some of the particles are stained black (possibly from iron sulfide), indicative of a reducing environment).

Lighter gray (2.5Y 4/1), slightly graded, well-sorted fine-grained silt. The base of this unit is composed of silt of a narrow size range, and only trace amounts of degraded organic matter and diatoms. The silt becomes less-well sorted upward (broader size distribution), and the clastic grains are angular and include trace amounts of volcanic glass and watershed rock minerals (primarily amphibolites).

"Load" structures observed at basal contact. Sediment is composed of ~20-25% organics, primarily degraded (which appear clumped). The diatoms, particulate organics, and clastic particles are finer-grained compared to the sample below.

Sediment is composed of large organic particles (some larger than 1 mm) and a smaller percentage (~10-15%) of clumped degraded organics, a wide variety of large diatoms, and broad range of clastic particles (in both size and composition).

recovery: normal background

post-disturbance: lake ecology disrupted

**Tail:** upward increase in % organic matter

med. silt slight grading

basal silt, becoming finer-grained upward

upward-fining, of both organics and clastics.

slightly denser layer

Normal background

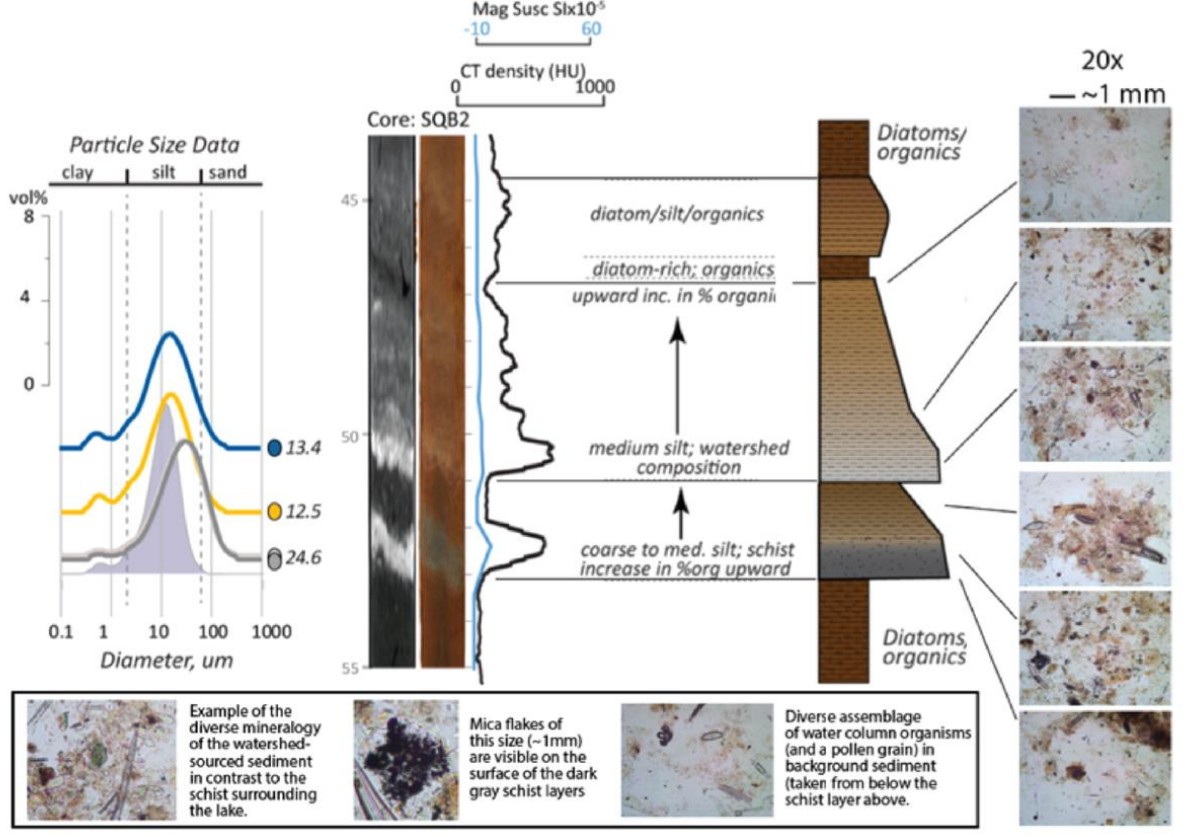

Mag Susc SIx10⁻⁵

CT density (HU)

Core: SQB2

**Particle Size Data**

clay | silt | sand

vol%

Diameter, um

13.4
12.5
24.6

Diatoms/organics

diatom/silt/organics

diatom-rich; organics
upward inc. in % organic

medium silt; watershed composition

coarse to med. silt; schist increase in %org upward

Diatoms, organics

20x
— ~1 mm

Example of the diverse mineralogy of the watershed-sourced sediment in contrast to the schist surrounding the lake.

Mica flakes of this size (~1mm) are visible on the surface of the dark gray schist layers

Diverse assemblage of water column organisms (and a pollen grain) in background sediment (taken from below the schist layer above.

Figure 3 (previous page). *Characteristics of earthquake-triggered deposits, as described in Morey et al., 2024 (this volume).* A. Type 1 earthquake deposit, attributed to the 1700 CE Cascadia earthquake, has load structures below the deposit base, followed by a fine-grained, well-sorted silt layer sourced from the Slickear Creek watershed (indicated by the presence of calcium minerals), followed by a long, organic-rich tail. There is evidence of a possible bypass turbidite below the base of the loading silt. This deposit sequence was attributed to the 1700 CE Cascadia earthquake. B. Type 2 earthquake deposit is a turbidite composed of lake-margin-sourced schist (represented by the lower schist deposit in this sequence; deposit I). This deposit sequence was attributed to the 1873 CE earthquake deposit, interpreted to be the result of a crustal earthquake sequence.

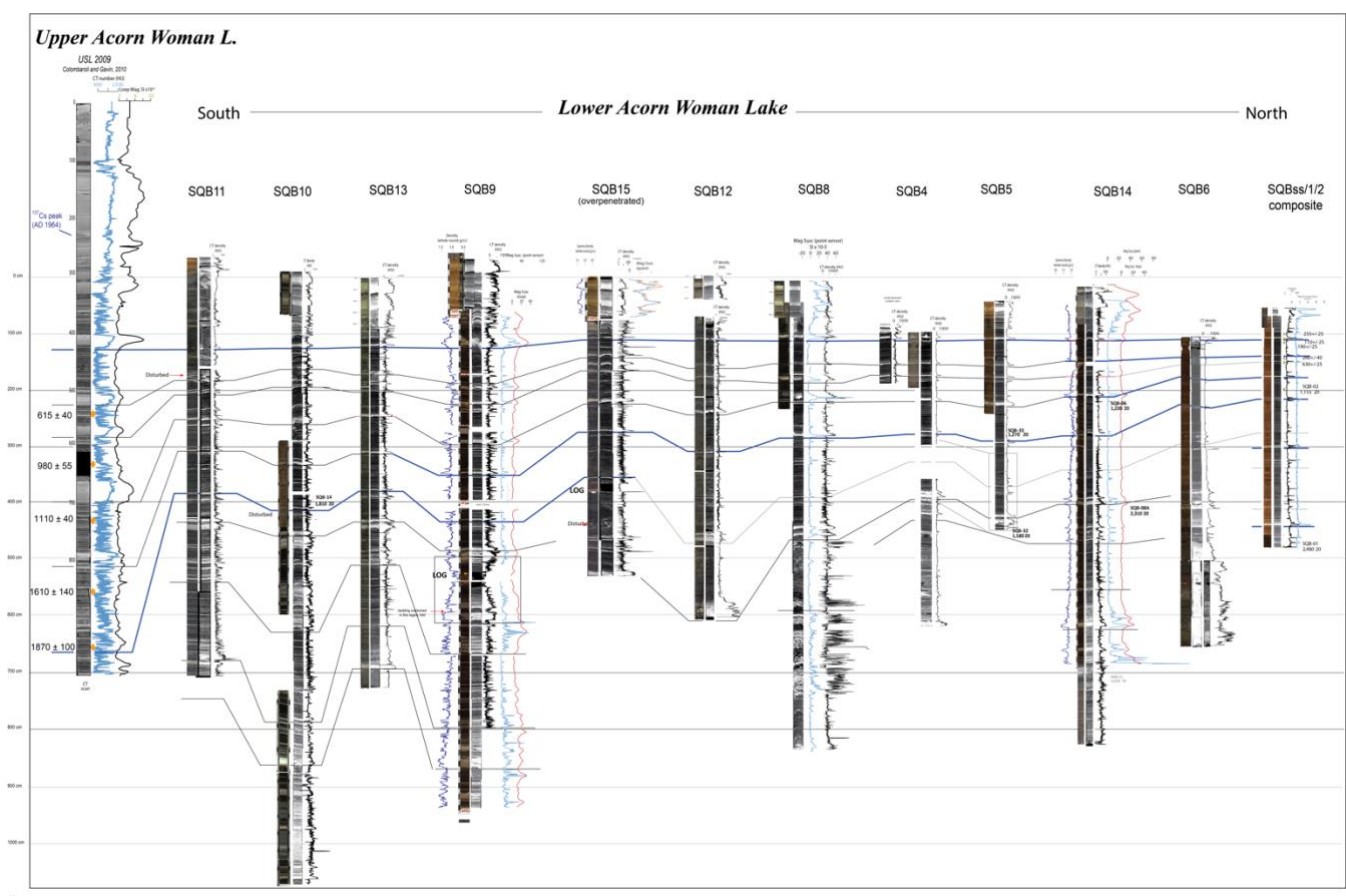

690

Figure 4. *Correlation diagram for all cores in Lower Acorn Woman Lake and relationship to the Upper Acorn Woman Lake core*. Cores are hung on the lake-wide disturbance deposit J, suggested in the companion manuscript (Morey et al., 2024, this volume) to be the result of the 1700 CE Cascadia earthquake. The thick line connects a lake-wide deposit that occurs in all cores that was deposited at ~1500 BP.

695

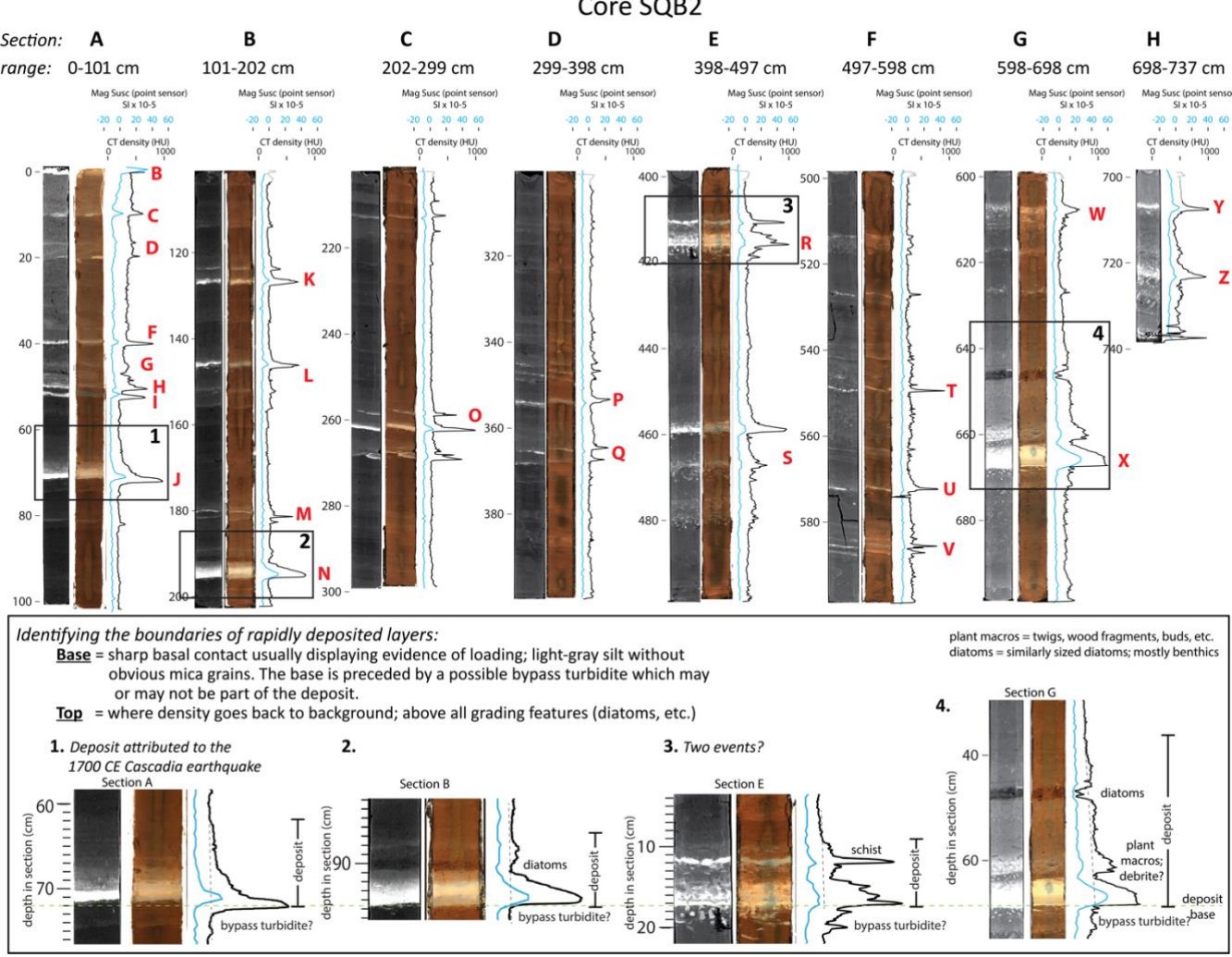

Figure 5. *Sections A-H, core SQB2*. Archival depths are in cm below the core top. Physical property excursions identified by red letters are disturbance deposits that are evaluated in this manuscript. Those identified by numbered boxes illustrate the complexity and variability in the expression of these disturbance deposits downcore.

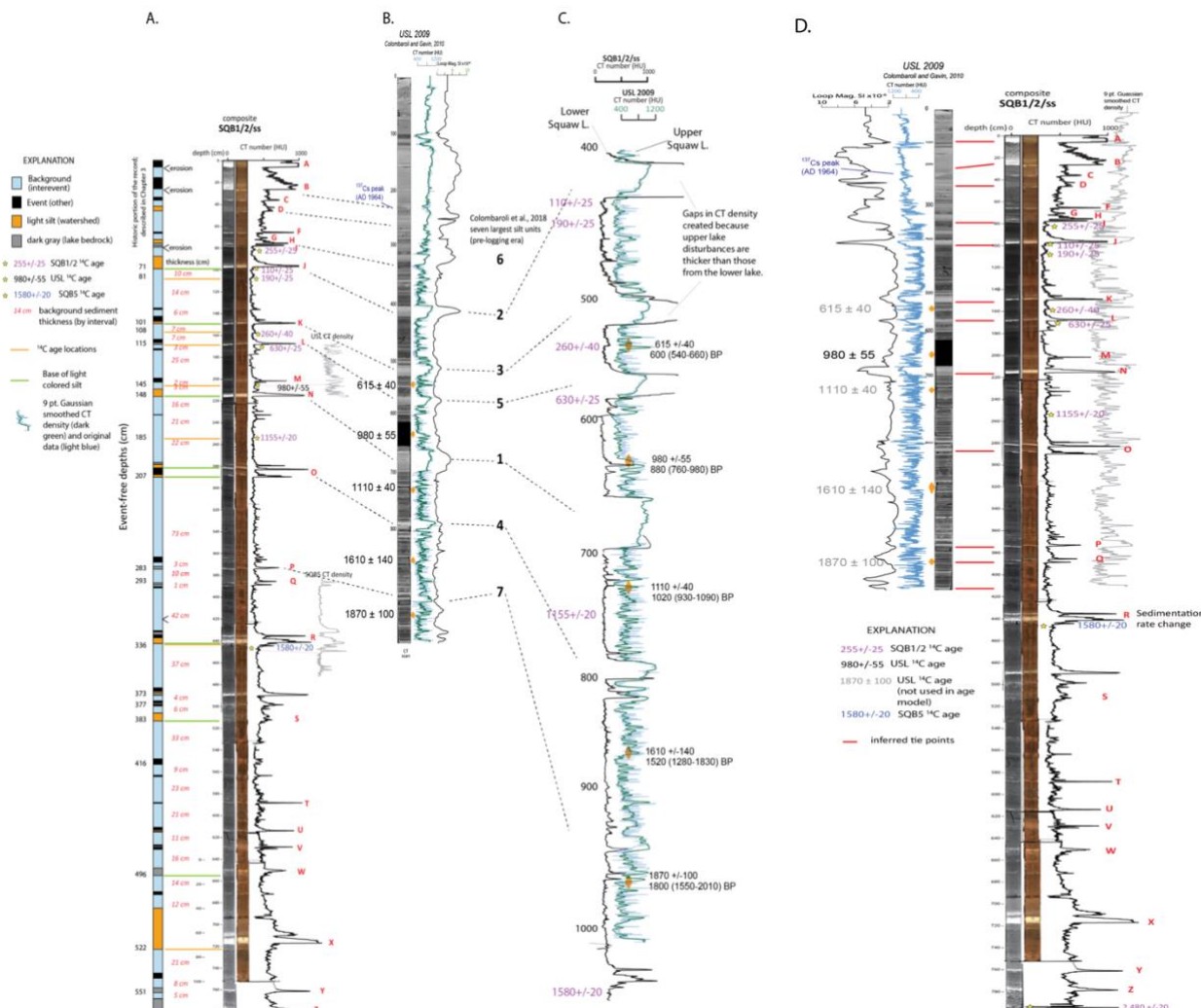

Figure 6 (previous page). *Identification of disturbance deposits and correlation between upper and lower lake records*. A. Red numbers represent the interevent thicknesses used in the event-free age-depth model. The red capital letters A-Z indicate the disturbances identified in this study. Grey traces to the right identify correlative sequences where age data have been used to supplement those radiocarbon determinations in core SQB1/2/ss. B. The relationship to the 2009 Upper Acorn Woman Lake core is shown by correlation lines (dashed). C. The relationship between the radiodensity data from core SQB1/2/ss (black trace) is shown compared to the radiodensity data from the upper lake core (green trace; 9 point Gaussian smoothing is shown over original data in blue). The relationships to the seven thickest deposits in the upper lake record compared to the lower lake record identified in Colombaroli et al., 2018 are identified by the dashed lines connecting numbers to events in the sequence. Note that the depth scale for the USL core (radiodensity units shown in blue) are true, but the depth of the lower lake core (radiodensity units shown in black) is not shown because depths have been distorted to match events. This is called flattening. Breaks in the lower lake radiodensity data were made in the middle of each thick deposit because the thicknesses of the upper lake deposits are much greater than the thicknesses of the lower lake deposits. Note that ages with +/- are radiocarbon determinations, and those with ranges in parentheses are calendar ages. See the Explanation for details. D. USL 2009 (left) was flattened to core SQB1/2/ss (right) to demonstrate the similarities between the core data. Flattening is a method whereby all the core data are transformed to match correlative horizons, in this case, correlative deposit bases. Correlated bases are identified by the red tie lines between cores. The correlation suggests that the radiocarbon ages identified in grey are older than the radiocarbon data would suggest for the lower lake core. Note that whole round magnetic susceptibility is in black and radiodensity is in blue (for core USL 2009) and radiodensity is in black for the core SQB1/2/ss. The grey trace to the far right is the USL 2009 smoothed radiodensity (9 point Gaussian window) to better compare the records (because the data in the upper lake core contains many more silt layers than the lower lake core).

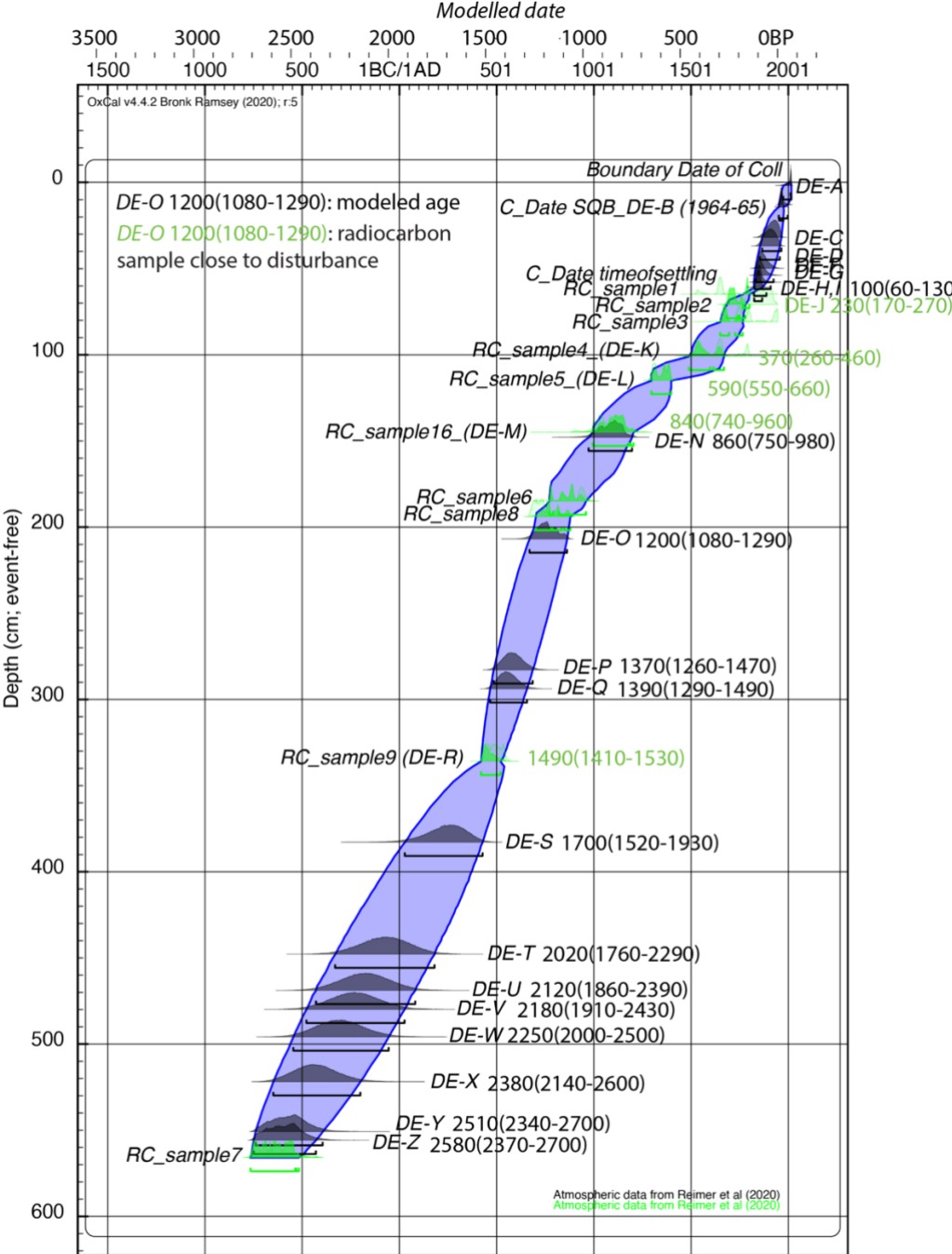

Figure 7. *Downcore age-depth model for Lower Acorn Woman Lake composite core SQBss/1/2*. Sample numbers (refer to Table 2) are positioned adjacent to their distributions. The envelope reflects the uncertainty (95% confidence) of the age-depth curve. Calendar ages in black are modelled ages and those in green are modelled ages by radiocarbon samples which are in close proximity to a disturbance deposit.

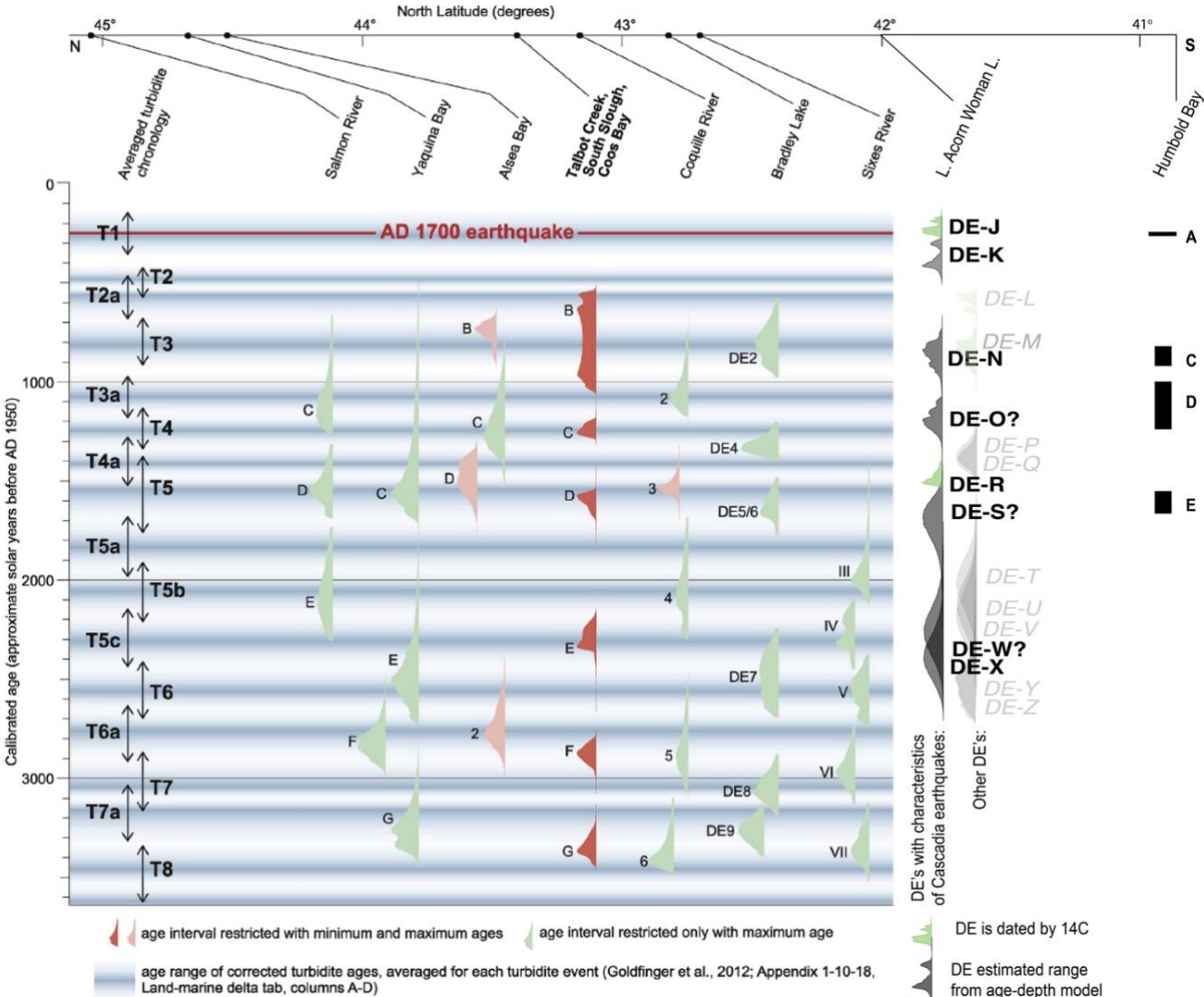

Figure 8. *Comparison of the Lower Acorn Woman Lake earthquake chronology to the compilation of southern Cascadia paleoseismic records by Milker et al., 2016.* At the far left are the marine age ranges of corrected turbidite, margin-wide averages (corrected for reservoir age) from Goldfinger 2012. At the far right are the disturbance deposits distributions for deposits K, N, O, R, S, W and X which are most similar to deposit J. Those distributions in green are deposits that have been directly dated. The other distributions in lighter grey are the remaining disturbances in the sequence that have other characteristics (schist-derived turbidites and thinner, less distinct, layers). Also to the far right are the results from Humboldt Bay; Padgett et al., 2021.

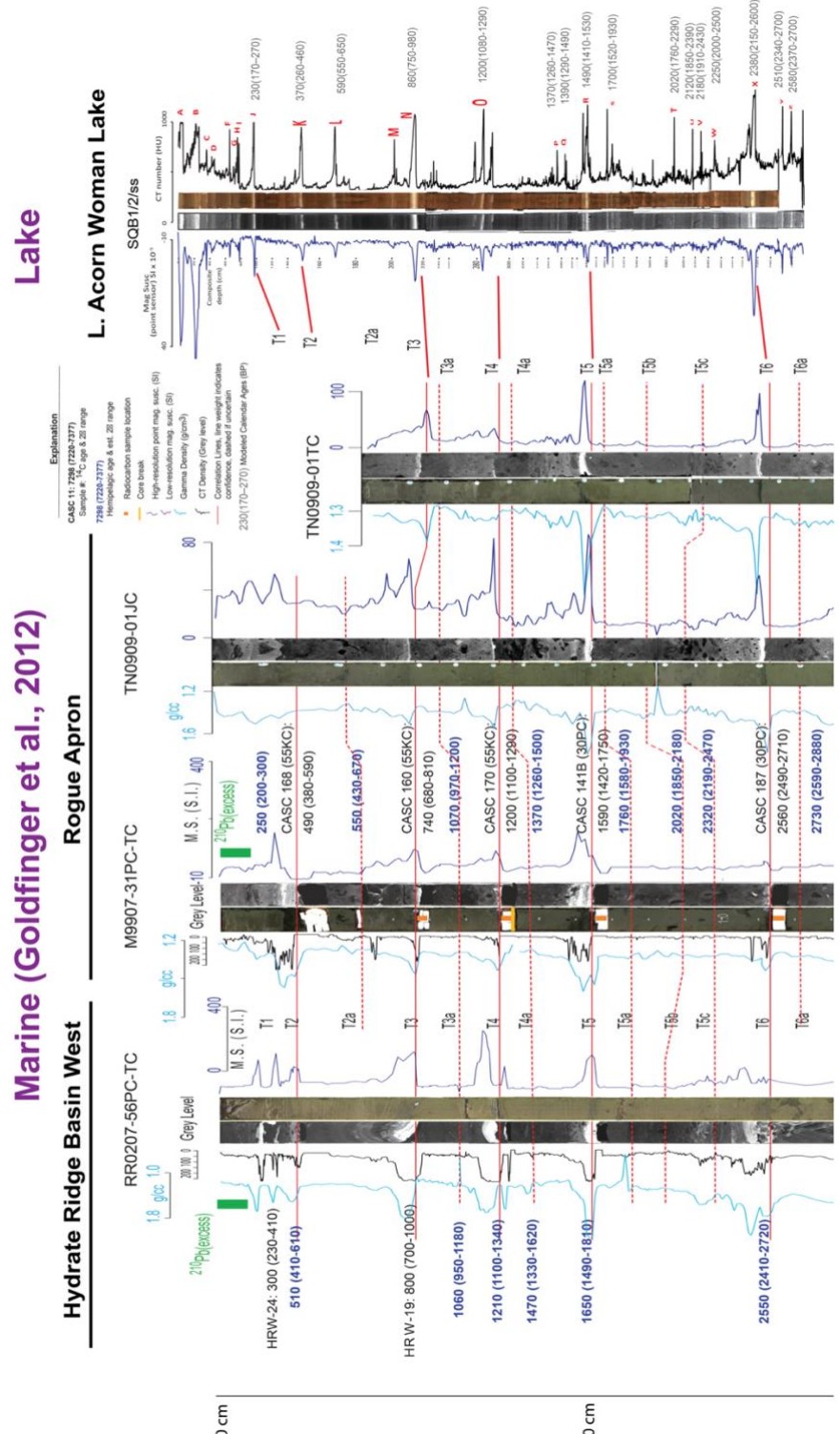

Figure 9 (previous page). *Marine-lake correlation diagram*. This diagram shows bed relationships for correlative units between Lower Acorn Woman Lake and marine paleoseismic sites Rogue Apron and Hydrate Ridge Basin West (Goldfinger et al., 2012). See Figure 1 for core locations.

750