# Peer review of "A 2700-yr record of Cascadia megathrust and crustal/slab earthquakes from Acorn Woman Lakes, Oregon"

_EGUsphere, 2023_

## Referee Comment (RC1)

Review: A 2700-yr record of Cascadia megathrust and crustal/ slab earthquakes from Upper and Lower Squaw Lakes, Oregon

By Ann Morey and Chris Goldfinger

The manuscript presents important paleoseimic data from a lacustrine archive at the southern end of the Cascadia subduction zone, where records have been scarce compared to the northern part. It also contributes knowledge to the characterization of so-called (disturbance) event deposits in organic-rich lake sediments where the identification of events can be difficult due to the homogeneous nature of such deposits as well as other factors such as low sedimentation rates etc. The amount of data presented is sufficient to support the main messages of the paper.

Despite the clear scientific merit, the concept used to identify earthquake-induced deposits and how these are generated remains somewhat unclear. There are also weaknesses in the background information and the structure of the paper. I believe these can be addressed but require some revision. Here are my major concerns:

1) It looks like that there is a companion paper to this paper characterizing the 1700 deposit in greater detail. Even if references are made, I believe that in order for this paper to stand alone, it must include more background information. I am wondering if the authors have considered to merge both papers?

2)  The seismicity of the southern Cascadia subduction zone has to be mentioned. For example, it is important to know the estimated groundshaking at the lake site for the historic earthquake in 1873 as well as the 1700 earthquake (or other large megathrust earthquakes).

3) More importantly, the geologic setting of the lake must be better characterized to understand the concept that is used to distinguish between large megathrust and smaller earthquakes. This includes the geology and geomorphology in the watershed of the lake as well as lake basin geometry and any delta or landslide deposits within the lake that could be destabilized during ground-shaking. This information is crucial to understand the source of the sediments found in type 1 deposits in contrast to the source of mineralogies found in type II deposits.

4) While the two types of deposits are relatively well characterized, I think that the processes that lead to the formation of these two types of event deposits remain somewhat uncertain. For example, I am having difficulties understanding what is meant by the watershed sourced turbidites. Do these turbidites incorporate sediment that comes from steep slopes in the surrounding watershed destabilized during ground shaking and subsequently transported into the lake (similar to NZ lakes in Howarth et al. 2014)? Is the other type of turbidite generated from slope failures within the lake? Is the difference of the two types just an effect of the amount of groundshaking at the site? I think I am not clear how you can distinguish between the different earthquake sources.

5) I think that it is not necessary to correlate physical proxies of other lake studies as shown in figure 9 and 10. I would merge these records with figure 8 or include a similar figure that just shows the correlation of the ages for those sites. I think this will also streamline the discussion section.

Minor comments:

**Abstract:**

I would avoid specific deposit names such as deposit J, turbidite T1, T2 etc. in the abstract and the introduction since they have not been introduced, yet. Consider some rephrasing and add a sentence on the methods used.

**Introduction:**

Maybe include short review on how earthquake induced event deposits look like and what other lake studies have found in the area. I am wondering about overlap with the referenced paper Morey et al. 2023. Can the two papers be merged?

**Setting:**

The first paragraph under "Methods" seems to be geologic setting. As mentioned above, this has to be expanded significantly regarding seismicity of the area and geologic setting of the lake.

**Methods:**

The title of the manuscript implies that there is also a record from Upper Squaw Lake. However, the methods only describe cores taken from Lower Squaw Lake. Has data from Upper Squaw Lake already been published?

Mention also XRD measurements that are mentioned later in the text.

I believe the last three paragraphs of the methods section can be shortened and possibly restructured.

**Inferred characteristics for earthquake types**

See my major comment 4. It is not clear what is meant by "Sediment sourced from watershed" and "Turbidite sourced from lake margin bedrock".

Yes, the structures at the base look like load structures. The question is, if these formed due to rapid sedimentation at the time of turbidite deposition or subsequent seismic loading. Maybe you can disucss later?

For both types I am missing a short statement about the lateral, lake basin wide distribution.

The list of characteristics at the end of this section seems to be a repetition.

**Results**

3.1: I would avoid the term "schist layers"

It seems as if the last two paragraphs that talk about correlation to Upper Squaw Lake should be under 3.2. I think the results section in general could be restructured a bit.

**Discussion**

The process of earthquake disturbance layers has to be described in more detail.

"Physical property peaks" is a little too vague.

I think the post-fire and flood-related erosional events can be shortened especially since fires are already excluded as a trigger in Upper Squaw Lake sediments.

I would leave section 4.2.2 and Figure 9 out. It is tricky to correlate selected physical proxies from two very different environments.

4.3: There are some references and terms in this section that don't seem related to the paper.

Section 4.4. is based on figure 9 and 10 which I would leave out and just add the age distributions to figure 8. See my comment 5.

From the manuscript it is not clear how the summary of ideal lake characteristics was established. Under Summary and Conclusions there is another paragraph that talks about the suitability of regional lakes. Consider merging/ rephrasing.

**References** not checked – sorry!

Figures: I am not sure if I have access to the highest resolution possible for these figures. Some seem fuzzy and are hard to read.

Figure 1: Could it be cut above northernmost location mentioned in text (I believe above ~45 deg N). Could you add a smaller overview figure showing the entire Cascadia subduction zone/ northwest Pacific and mark extent of southern Cascadia?

Figure 2: I have a hard time reading the core names but it looks like this study is mostly based on cores from the lake side and not from the deep basin? It is important to explain why those were chosen since deep basin cores would probably show a more complete record.

Figure 3: What is meant by calcium minerals? If $CaCO_3$ data exits from LOI it might be useful to plot here especially if it correlates with a specific source area for the turbidites. I think the figure can be simplified. Not all smear slide pictures and detailed descriptions are necessary.

Figure 4: very hard to see details. I suggest plotting only selected cores at a larger scale.

Figure 6: I don't think D is necessary. Maybe merge information with A, B and C and enlarge.

Figure 8: Is it possible to add ages of events from Figure 9 and 10 here? Also, can you indicate what archive each site represents (marsh record, lake, offshore, etc.)

---

## Referee Comment (RC2)

**A 2700-yr record of Cascadia megathrust and crustal/slab earthquakes from Upper and Lower Squaw Lakes, Oregon**

Ann E. Morey ( ✉ ann@cascadiapaleo.org )

Cascadia Paleo Investigations    https://orcid.org/0000-0002-8702-2581

Chris Goldfinger

Oregon State University    https://orcid.org/0000-0002-4603-6178
* * *
Research Article

Keywords: Cascadia subduction zone, lacustrine paleoseismology, subduction earthquakes

Posted Date: July 17th, 2023

DOI: https://doi.org/10.21203/rs.3.rs-2277419/v2
* * *
**Abstract**

We infer a ~ 2,700-year history of Cascadia megathrust and other earthquakes from two small mountain lakes located 100 km inland of the coast near the California/Oregon border. We use the characteristics of disturbance deposits in the historic portion of the sediment cores from the lower lake to identify a deposit from the 1700 CE Cascadia earthquake (deposit J). This deposit is composed of light-coloured silt (indicating it is enriched in watershed-sourced sediment), without visible mica grains (which would indicate a lake bedrock source), organic grading of the deposit tail, and a basal contact with evidence of rapid loading.

Seven deposits downcore have some of the characteristics of deposit J. An age-depth model suggests that the five deposits most similar to deposit J (including deposit J) are temporal correlatives to the largest margin-wide marine turbidite event deposits from Goldfinger et al., 2012, (T1, T2, T3, T4, T5 and T6), whereas the two deposits with some of the characteristics are potential correlatives of smaller turbidites T5a and T5b. Other thinner deposits are temporal correlatives of T2a and T3a and other smaller deposits of uncertain origin. Lake core physical property data can be correlated to those from other regional lake records and offshore cores. These results suggest that small Cascadia lakes with sufficient sedimentation rates (~ 1−2 cm/decade) with mixed clastic and organic sedimentation may be good recorders of earthquakes, that subduction earthquake deposits are different from those from other types of earthquake deposits and deposits from other types of disturbances, such as floods.

**1 Introduction**

[revised manuscript text omitted]

We identified disturbance deposits in cores as abrupt increases in CT density data, as described in Morey et al. (2013). These disturbance deposits were then correlated throughout Lower Squaw Lake using physical property data using modified well-log techniques (Fukuma, 1998; Karlin et al., 2004; Abdeldayem et al., 2004; Hagstrum et al., 2004; Waldmann et al., 2011; Goldfinger et al., 2012; Patton et al., 2015) for cores constrained temporally using ages from detrital macrofossils. Traditional well-log correlations use the physical property data from drilled wells to identify and trace lithologic units. Similarly, this project uses magnetic susceptibility and density data for correlation. This has the benefit of radiocarbon age control data but does not use gamma density or p-wave velocity. These sediment physical property data allow deposits to be correlated based on deposit composition and structure (Amy & Talling, 2006) and is widely used to correlate seismogenic marine turbidites.

We sampled the Lower Squaw Lake cores for radiocarbon after splitting the cores longitudinally. Fragile detrital plant macrofossils sampled from targeted horizons of undisturbed sediment, were cleaned and dried, then analysed by AMS (accelerator mass spectrometer) for radiocarbon. We selected the target horizons for sampling based on a tentative relationship between the dated sequence from Upper Squaw Lake (Colombaroli & Gavin, 2010) and the Lower Squaw Lake stratigraphy. We did not acquire [210]Pb and [137]Cs data for Lower Squaw Lake cores to get historic sedimentation rate data because the upper portions of the sediment cores contain two thick clastic units (found lake-wide and of varying thickness) with evidence of erosion at the basal contact, which violates the assumptions of these dating methods of continuous sedimentation required to create a sedimentation rate curve. We used the correlation of physical property data, in particular magnetic susceptibility and CT density, between the upper and lower lakes to infer that the younger of these disturbances was deposited in 1964 (as presented in the supplementary data).

An age-depth model for the historic portion of the record was developed from an event-free sequence (e.g., Enkin et al., 2013; Hamilton et al., 2015; Goldfinger et al., 2017) using CT density. The base of each disturbance deposit was determined to be the location where CT density rapidly increases from background sediment and the top of the deposit was determined to be where CT density drops below background levels. Disturbance deposits without evidence of inter-event sedimentation are treated as a single disturbance deposit for the purpose of the age model. Disturbance deposits show significant variability downcore, which complicates the boundary identifications (described in more detail below). A final age-depth model was created using a P_sequence in the Bayesian software OxCal (Bronk-Ramsey, 2017).

**2.2 Inferred characteristics for earthquake types**

Evidence presented in Morey et al., 2023 (this volume), suggests that there are two types of earthquake-generated disturbance deposits, and that both are different from the deposits of other types of disturbances (i.e., flood deposits) in the historic portion of the record from Lower Squaw Lake. Deposits attributed to earthquakes have organic rich tail deposits, dense silt (sourced from the watershed) at the base and show evidence of loading (detailed below) where the silt lies on the organic-rich sediment below the deposit. Although evidence of loading is not definitive evidence of earthquakes, it is part of the suite of criteria used based on characteristics of deposit J. The types of earthquake-generated disturbance deposits identified and described in Morey et al., 2023 (this volume) are presented below.

Type 1 (Figure 3a). Well-sorted medium silt sourced from the watershed with an organic tail. Example: deposit J, inferred by Morey et al., 2023 (this volume), to result from the ~M9 1700 CE Cascadia earthquake. Deposit J is a thick (~7-15 cm), dense (~1,000 HU at the base), weakly graded, medium to fine-grained silt unit with an organic-rich tail. The base is composed of fine-grained, well-sorted silt (~90% inorganics) that appears "clean" (lacking other components such as broken diatoms and organic matter). The layer of silt is 1.5-4.0 cm thick (depending on location in the lake) and becomes less-well-sorted upward with grading. As grading proceeds upward, the silt becomes more fine-grained, and the organic content gradually increases upward. The particle-size distribution at the base of the deposit is narrower than the rest of the disturbance (as shown in Figures 3a and 3b) and pure (predominantly silt with only trace amounts of diatoms and organic particles). This fine-grained (medium-fine silt) sediment was interpreted to sink into the less-dense sediment below, which we interpret as the result of loading ("LOAD" in Figure 3a). X-ray diffraction (XRD) demonstrates that this silt (sampled from core SQB2A) is composed primarily of sediment sourced from the watershed. Above the deposit tail, background sediment contains a different suite of diatoms compared to prior to the earthquake after the tail deposit suggesting a post-earthquake change in community structure (the types and relative abundance of species living in the water column). It was inferred that the processes triggered by the earthquake removed organisms during settling, altering the types of organisms present in the water column.

Type 2 (Figure 3b). Lake-wide turbidite sourced from lake-margin bedrock. Example: Deposit I from the deposit sequence H/I inferred to result from the ~M7 1873 CE earthquake. The lower portion of the deposit sequence (deposit I) is dark gray (GLEY2 4/5PB) coarse silt dominated by large, visible mica flakes (~90% inorganic). The deposit has a sharp basal contact and is initially reverse, then normally, graded. The reverse-graded portion of the deposit contains a higher percentage of organics (including rootlets) compared to the normally graded portion of the deposit. The base of the deposit is very sharp with evidence of erosion. This deposit is a turbidite deposit which was identified by XRD to be composed of schist (evidence that it is sourced from the lake margin). It is unknown if deposits H and I are the result of the same or separate events.

For this study, we used the distinctive characteristics of deposit J to identify other potential Cascadia earthquake deposits downcore. These characteristics are:

1. Light-coloured (Munsell colour: 2.5Y 4/1, indicating a watershed source), well-sorted silt without visible mica grains.

2. Evidence of loading (projections of silt into the organic sediment below) of the basal silt into the organic-rich sediment below.

3. Presence of a long (typically 2-5 cm but varies with location in the lake) organic-rich tail.

We then used the characteristics of deposit I, inferred to be the result of the 1873 CE Brookings earthquake, to identify other types of earthquakes in the downcore record. This is a lake-wide turbidite composed of dark grey (Munsell colour: GLEY2 4/5 PB, indicating a lake bedrock source) schist containing visible mica flakes and a short (~1 cm) deposit tail.

**3 Results**

**3.1 Identification of earthquake-generated disturbance deposits in Lower Squaw Lake**

Disturbance event deposits were identified in the sediment cores used in this study (Table 1) as described in the methods section, then expressed as a correlation diagram (Figure 4). Distinctive beds were correlated using the age data, sedimentology, and physical property data (described further below). Shallow water cores and cores from the northern portion of the lake have less sediment between time equivalent horizons than do the deep water and southern lake cores. The deep-water cores contain thicker disturbance deposits and contain slumps and folds, and occasionally woody debris and portions of soft, partially degraded logs associated with the slumps and folds (labelled on the diagram). A core from the northern site (composite core SQB1/2/ss) was selected to create the chronology to avoid these disturbances in the southern lake cores, avoid an influence from the landslide to the south, and reduce the influence from the second watershed (via Squaw Creek).

Deposits suspected of being triggered by Cascadia earthquakes were identified in the composite core SQB1/2/ss using the characteristics of deposit J as described in the Methods section. Five deposits (deposits K, N, O, R, and X; see Figure 5) were identified as most similar to deposit J in the downcore record. These disturbance deposits are all light-coloured silt layers with few or no visible mica grains, show some evidence of loading into the organic sediment below, and have organic-rich tails. Deposits K and N in the upper portion of the core (Figure 5, bottom left) are most similar to deposit J. These deposits have a well-sorted medium silt that bleeds (fine-grained loading) into the organic sediment below. Deposits O, R and X also have similar characteristics, but are slightly different from, deposit J. For example, deposit O is a thin, light coloured silt with little evidence of loading, deposit R contains evidence of loading as medium silt finger-like projections that are broken off, and deposit X is a large (30 cm long) turbidite showing unusual grading characteristics (including layers of plant macrofossils and benthic

diatoms). None of the labelled deposits has a tail as long as deposit J (the closest is deposit H, which precedes deposit J). These details can be seen in core SQB2 as shown 
[revised manuscript text omitted]

Sediment and water are required to create earthquake-triggered disturbance deposits in lakes. Water transports sediment to the lake and the sediment is deposited, loading the lake margins. Whereas water supply is a function of climate, vegetation, substrate and watershed size and characteristics, sediment supply can fluctuate as a result of changes in landcover (from changes in climate or as a result of wildfires), or as a result of human activities such as logging and road building. Likewise, with a sufficient supply of sediment and water, sediment can also be transported to the lake to create flood-induced disturbances.

Physical property peaks in lake sediment cores can be the result of a variety of events. These include 1) aseismic events such as post-fire erosion, storm remobilization of sediment, land-use changes and flooding, 2) seismic events which trigger landslides and submarine lake floor sediments, 3) mixed seismic and aseismic events, such as the destabilization and subsequent transport of destabilized hillslope sediment into the lake, 4) concentrations of authigenic minerals such as magnetite, 5) shaking from nearby volcanic activity, and 6) layers of volcanic tephra. This study seeks to differentiate between plate boundary earthquake, non-plate boundary earthquakes, and aseismic disturbance deposits triggered in-lake disturbances by comparing the record of disturbances from Squaw Lakes to published records of

Cascadia megathrust earthquakes. First, however, we determine if the beds are internally (within-lake) or externally (watershed) sourced.

**4.1.1 Post-fire and flood-related erosional events**

Lakes throughout Cascadia have been used extensively to reconstruct fire histories and post-fire erosion, where minerogenic layers and charcoal abundance data are frequently used to infer increased erosion after large wildfires (e.g., Millspaugh and Whitlock, 1995; Colombaroli and Gavin, 2010). Although charcoal analysis has been successfully used as a proxy for post-fire erosional events, Long et al. (1998) show that peaks in charcoal accumulation do not always correlate with magnetic susceptibility at Little Lake, Oregon. Similarly, charcoal peaks at Bolan Lake, Oregon, and Sanger Lake, California, (between the coast and Squaw Lakes at a similar latitude) are not always associated with (or immediately precede) magnetic susceptibility peaks (C. Briles, pers. comm., 2010).

Although wildfires are common in Cascadia and charcoal analysis at nearby Upper Squaw Lake suggests that wildfires do influence erosion in this region (Colombaroli and Gavin, 2010), subsequent analysis of the pseudo-annual silt layers observed in the CT data has suggested that the seven largest disturbance events in the record (pre-logging era) are the result of a different process at Upper Squaw Lake other than erosion, possibly earthquakes (Colombaroli et al., 2018). The study demonstrates that although the rate of silt deposition (mm/year) displays a power law distribution, there are seven largest silt layers that are thicker than expected, suggesting a different process controls the silt accumulation during those times.

Runoff-initiated post-fire subaerial debris flows have been observed to be a significant erosional process in the Western Cascades, Oregon (Wall et al., 2020). These debris flows could create hyperpycnal flows resulting in thick debris flow deposits (such as turbidites and hyperpycnites) and must be considered as a possible origin for the seven largest deposits in the Upper Squaw Lake record. This is considered unlikely, however, because the correlative units in the Lower Squaw Lake record do not show evidence of basal erosion (coarse basal sediment and entrained organic matter) as would be expected for a debris flow deposit. For example, a comparison of the basal layer of deposit B (assumed to be a debris flow turbidite) and deposit J (attributed to the 1700 CE Cascadia earthquake) are different in the following ways. Whereas both basal silt layers have roughly the same amount of clastic material (~90% by weight) their compositions are completely different. The base of deposit B is composed of poorly sorted coarse-grained silt and contains degraded and particulate organic matter including rootlets which affects the magnetic susceptibility of the lower half of the deposit. In contrast, the base of deposit J is composed of very well-sorted medium-grained silt and contains very little degraded organics, few visible particulate organics, and magnetic susceptibility is high throughout the silt. Although the grain-size of the deposits similar to deposit J increase in grain-size (to coarse silt) downcore, the other characteristics of the deposit are very similar (high magnetic susceptibility, less degraded and particulate organics, and narrow particle size range).

Both floods and post-fire erosion, however, may be a source for the smaller deposits in the Lower Squaw Lake record other than those that are correlated to the seven thickest deposits in the Upper Squaw Lake record. Flood disturbances have been identified in the Lower Squaw Lake record (presented and discussed in chapter 3) as deposits with slight increasing, then decreasing, grain size and % clastics with a physical property profile that is somewhat rounded in appearance, typical of the waxing and waning of storm events (Mulder, 2001). This is also supported by data from St-Onge et al. (2004) who analysed a sedimentary sequence from Saguenay Fjord, Québec, that was produced from a known historic earthquake followed by a flood (landslide dam breach). The analysis showed that the hyperpycnite deposit resulting from the flood has reverse, then normal, grading whereas the earthquake-triggered turbidite deposit has normal grading.

At Upper Squaw Lake, silt accumulation was strongly correlated with precipitation during the logging era (1930-present), however prior to that time the occurrence of the thicker silt layers (other than the seven thickest) was related to fire (determined because they were preceded by high charcoal concentrations). This suggests that at this location (Squaw Lakes) flood deposits are not the thickest silt units and where present in the lower lake, even the largest historic flood layers appear to have a rounded appearance in their physical property data (see for example, deposit G and the deposit suspected to be the result of the extreme flood from the 1861-62 atmospheric river event flood just below deposits H and I).

**4.1.2 Earthquakes**

Morey et al., 2023 (this volume), suggests that plate boundary, intraplate and/or crustal earthquakes influence the record at Squaw Lakes. To evaluate this, we first use what is known from the historic portion of the record to suggest interpretations based on the sedimentology, then we compare the temporal relationship between the Lower Squaw Lake record and nearby paleoseismic records of Cascadia earthquakes.

**4.2 Deposits similar to Deposit J**

**4.2.1 Temporal relationship to nearby paleoseismic events (marine and coastal)**

Figure 6a identified the 26 disturbance deposits, DE's A-Z, that have large excursions in magnetic susceptibility in the downcore record from Lower Squaw Lake. Nine of these disturbance event deposits (Table 3; including deposit J and deposit H) have some of the characteristics of deposit J, which was suggested in Morey et al., 2023 (this volume), to have formed in response to ground motions from the 1700 CE Cascadia earthquake. The other disturbance deposits identified in the downcore record are also of higher density and magnetic susceptibility compared to background sediment, but do not have a distinctive watershed sourced composition or some of the other characteristics of deposit J.

Table 4 identifies the beds from Lower Squaw Lake and possible correlatives from marine and coastal paleoseismic data– the T numbers in bold are the thickest beds at Rogue Apron site which are also those deposits that correlate to beds at Hydrate Ridge West (Figure 8; both of which are marine turbidite sites from Goldfinger et al., 2012). This suggests that the disturbances with watershed-sourced silt, organic tails, and evidence of loading from the Lower Squaw Lake record are most likely the result of significant plate boundary earthquakes (identified in Goldfinger et al., 2012, and summarized in Walton & Staisch et al., 2021). These characteristics are consistent with, but not exclusive to, earthquake triggered events.

Table 4 indicates that there is an excellent temporal match between the four coastal, lake and marine paleoseismic sites for the margin-wide events T1, T2, T3, T4, and T5 (using the marine turbidite bed notation). T1/DE-J, T3/DE-N, and T4/DE-O correlatives have radiocarbon determinations which are within a few decades, whereas there is poorer agreement between T2/DE-K and T5/DE-R medians and ranges even though the ranges overlap significantly (some ranges are simply larger than others). Although DE-X is suspected to correlate to T6 (based on a comparison of physical property data between lake and marine cores), the age range for DE-X is significantly larger (460 yrs for DE-X compared to a few hundred years for Rogue and Hydrate Ridge sites) making this linkage less certain. A distinctive 1000 yr gap occurs in the record of Cascadia earthquakes between T5 and T6 in all records (Atwater 2004; Kelsey et al., 2005; Goldfinger et al., 2012; and Witter et al., 2012). This can be seen in the Lower Squaw Lake record as well. A graphical representation of the relationships between these age distributions can be seen in Figure 8.

**4.2.2 Correlation of physical property data between Lower Squaw Lake and the marine paleoseismic record**

The Lower Squaw Lake record was correlated to marine sites Rogue Apron and Hydrate Ridge Basin West (Goldfinger et al., 2012; see Figure 1 for core locations) using physical property and radiocarbon data. The relationships between cores are shown in the bed-flattened correlation diagram (Figure 9). This method of correlation appears to work even though the physical property data for the marine cores typically reflects the amount of magnetic minerals and grain size of horizons downcore, whereas the physical property data is also influenced by the percentage of organic matter (which can be part of the graded sequences) in the lake core.

**4.3 Smaller disturbance event beds**

There are smaller deposits in the Squaw Lake sequence that were not identified as potential Cascadia megathrust earthquake deposits because they do not have the characteristics of deposit J described in Morey et al., 2023 (this volume). Some of these may correlate to the large number of smaller deposits in the marine record. Deposit L is one example. This deposit is dark grey turbidite with visible mica flakes, and therefore is more similar to the wall failure deposit E and seismoturbidite deposit I described in Chapter 3. This deposit has a correlative unit in the Upper Squaw Lake record (dated to 550-670 BP, twosigma range, Figure 6) and is contemporaneous with marine event T2a. The different composition (schist) and characteristics (it is a turbidite with a short tail) of this deposit in the lake core suggests that it was not a Cascadia earthquake that triggered this deposit. It also does not match the timing of anything on the northern San Andreas fault (nor do T2b, T3a, or T4a; see Goldfinger et al., 2019, 2020). For these smaller events our uncertainty is higher because we can't use size of the event (because frequency is a function of distance), and there is more uncertainty in timing. These events, then, are harder to interpret. Lakes as well are not perfect recorders and therefore we cannot simply say an earthquake occurred or didn't occur simply based on the physical property data. Based on the available information, however, our preferred interpretation is that the thickest of these schist layers are the result of separate events and not complex deposits in response to shaking from an earthquake because they occur in various combinations (as a single deposit, and before and after the initiation of a Cascadia-like deposit) relative to Cascadia earthquakes. Without further information, because T2a, T2b, and T4a are not San Andreas fault earthquakes and they are present in the Upper and Lower Squaw Lakes records as well as at Rogue and other southern Cascadia sites, it is suggested that they may be the result of southern ruptures of the Cascadia fault.

Other sources of seismicity may also influence these records: earthquakes on crustal faults (such as the San Andreas fault or other unidentified regional faults), the Gorda Plate, nearby transform faults, and intraplate earthquakes. There have been suggestions that both the M7.9 1906 CE San Andreas earthquake and the ~M7.0 1873 CE Brookings intraplate earthquake are found in both the marine and lake records (Morey et al., 2023 (this volume), and Goldfinger, 2019; 2020). For example, other regional lakes, such as Bolan and Sanger Lakes, both potentially contain evidence of T2 and T2a, although the resolution of the Bolan Lake record has less-distinct deposits compared to Sanger Lake (see Figure 10). This work is ongoing and beyond the scope of the work presented in this article.

**4.4 Data integration**

The correlation between the Lower Squaw Lake disturbance record and Rogue Canyon and Hydrate Ridge marine sediment cores (Figure 9) suggests that all the full-margin ruptures (T1-T6), including T2 (using the marine turbidite T numbers from Goldfinger et al., 2012), younger than 2700 BP disturb sediments in Lower Squaw Lake. T2 is unusual because it is not found in any of the coastal southern Cascadia paleoseismic sites (Coquille River, Bradley Lake, and Sixes River; Kelsey et al., 2005; Witter et al., 2012) but is found in the marine turbidite record of Goldfinger et al. (2012; 2013). These southern coastal sites record tsunamis, and to have created a deposit at the site the tsunami must have been large enough to have entered the lake or estuary. Megathrust earthquakes that produced turbidite deposits T1, T3, T4, T5 and T6 caused large tsunamis at Bradley Lake (Figure 8).  It is possible that T2, a smaller turbidite, may have been triggered by a smaller earthquake which produced a smaller tsunami that did not overcome the threshold to leave a deposit at Bradley Lake or other coastal sites.

Although T3a, a southern Cascadia event, was not identified as an individual disturbance event deposit in the Lower Squaw Lake record, there is a disturbance event with a smaller magnetic susceptibility and CT density signature in the core. This suggests that even these smaller disturbance events in the Lower Squaw Lake record may be the result of Cascadia earthquakes.

The timing of T5 and T6 are similar to the timing of the formation of Lower Squaw Lake (2470-2700 BP; ~T6) and Upper Squaw Lake (1420-1530 BP; ~T5). Did shaking from a megathrust earthquake cause the landslide to fail, creating the lakes or are failures of the landslide the result of nearby crustal faults? High-frequency ground motion from a nearby intraslab earthquake is suspected in Morey et al., 2023 (this volume) to be the cause of the landslide dam failure in 1873 CE, suggesting that local earthquakes can disturb the landslide, however it is also possible that the landslide is unrelated to earthquakes. This is important because there have been several studies that have attempted to link landslides in the Coast Range and elsewhere to Cascadia earthquakes without success (see, for example, Struble et al., 2020 and LaHusen et al., 2020).

The correlation between physical property data between the Upper and Lower Squaw Lakes records and the Rogue Apron and Hydrate Ridge West marine seismoturbidite records (Figure 9) suggest that these sites are being influenced by the same events. Given this large-scale correlation over multiple depositional environments it would be expected that nearby lakes would also contain similar types of event deposits that are easy to correlate, however this is not so (Figure 10). There are several reasons for this. First, the lakes are very different from one another. Both Bolan and Taylor lakes are spring fed, with little clastic supply, resulting in disturbance events that are difficult to see without very high-resolution data and a careful examination of the sedimentology of the cores. Sanger Lake is slightly different in that it is a cirque lake with a large glacial rock fall upstream of the lake. Strong ground motion could cause settling of the large boulders and cobbles, resulting in the thick lake-wide clay layers interpreted to be the result of the earthquakes that caused the marine deposits T1 and possibly T4 (Figure 10). Why these earthquakes would produce these thick layers (and not the other large earthquakes), is unknown. Again, high resolution data, in particular CT data, colour imagery and high-resolution point magnetic susceptibility data, in conjunction with careful examination of the stratigraphic sequences is required to understand the sedimentary record from this site. Furthermore, it is imperative that the sediment cores be carefully evaluated prior to sampling for radiocarbon to avoid sampling from organic disturbance layers that are frequently formed in response to shaking in lakes dominated by organic sedimentation.

It is also possible that some lake types are more conducive to preserving evidence of earthquakes than others. Based on this study the following characteristics of lakes are suggested as optimal sites for paleoseismic studies in Cascadia:

1. Small lakes (< 1 km$^2$); landslide dammed lakes.

2. High sedimentation rates (~1-2 cm/decade); mixed (roughly 50% each) clastic and organic sedimentation.

3. The presence of a delta as a source of fine material for liquefaction.

4. Water depth greater than ~7 meters to prevent an influence from bioturbation.

**5 Summary and Conclusions**

Seven of the 26 disturbance event deposits identified in this chapter have the characteristics of deposit J, which has been attributed to the 1700 CE Cascadia megathrust earthquake. These disturbances correlate to the thickest disturbances in the Upper Squaw Lake record, and to full margin megathrust events onshore coastal and offshore marine turbidite records over the past 2700 years: T1, T2, T3, T4, T5 and possibly T6 (as interpreted by Goldfinger et al., 2012; other interpretations of the onshore and offshore data exist). There are smaller events in the Lower Squaw Lake record as well. T2 and T2a marine equivalents are missing in the tsunami record from Bradley Lake, however both have temporal equivalents present in Lower Squaw Lake. Most of these smaller events cannot be attributed to specific events, however there is the potential to sort this out in the future with further analysis and additional radiocarbon ages. For example, although T3a was not identified as a significant disturbance in the Lower Squaw Lake record, there is a smaller disturbance below T3 that could be dated to determine if it is contemporaneous with T3a in the offshore record.

Some of the smaller events are schist layers which appear to be independent responses to a single event not part of a complex as interpreted in Morey et al., (this volume). Table 4 shows that the Lower Squaw Lake record has the temporal equivalents of T2a, T4a, T5a, T5b and T5c. The only event missing in this time range is T3a.  Of these, T2a, T4a and T5b are interpreted as possible crustal events, while T5a and T5c equivalents are more similar to plate boundary earthquakes.  So, it seems possible that there are mixed sources for these smaller southern events, but it is not possible to tease them apart without more information.

The relationship between regional lakes needs further investigation into the sedimentology and fine-scale details of the stratigraphy to interpret, however there is evidence that regional cores also contain evidence of Cascadia earthquakes. The signal is not as strong, but this may be a result of the lack of high-resolution data, especially CT data, for these cores. The strength of the signal in these cores (Bolan and Sanger lakes) may be a result of the lack of clastic supply because they are primarily spring fed lakes. This suggests that landslide dammed lakes or those that are stream fed may be better sources because they load the lake margin with sediment and create deltas, both of which are conducive to producing disturbance event deposits in lakes.

**Declarations**

[revised manuscript text omitted]

**Figures**

[Figure]

**Figure 1**

*Location map.* The yellow stars identify the location the lakes presented in this study (California: BRL = Black Rock Lake, TL = Taylor Lake, CaL = Campbell Lake, ML = Muslatt Lake, SL = Sanger Lake; Oregon: Bolan Lake, HL = Hobart Lake, BdL = Bradley Lake, TrL = Triangle Lake). Other sites mentioned in the text are at Sixes River, just south of BdL, and Coos Bay, just north of BDL. volcanoes are identified by orange circles. The base map (adapted from Goldfinger et al., 2012) identifies the location of channel systems and sediment cores used to reconstruct the offshore record of Cascadia earthquakes.

[Figure]

**Figure 2**

Lake Setting. Upper and Lower Lakes are shown with core locations. Left: The lakes are connected hydrologically by a small stream (Squaw Creek) that crosses a portion of the landslide that created Upper Squaw Lake. Right: The core locations for each of the sediment cores from Lower Squaw Lake are identified by triangles (2015 cores) or X's (2013 and 2014; red). The Upper Squaw Lake core was taken from 14.1 m water depth, at the lake's depocenter.

[Figure]

**Figure 3**

*Characteristics of earthquake-triggered deposits, as described in Morey et al., submitted; this volume*. A. Type 1 earthquake deposit, attributed to the 1700 CE Cascadia earthquake, has load structures below the deposit base, followed by a fine-grained, well-sorted silt layer sourced from the watershed (indicated by the presence of calcium minerals), followed by a long, organic-rich tail. B. Type 2 earthquake deposit is a

turbidite composed of lake-margin-sourced schist (represented by the lower schist deposit in this sequence; deposit I). This deposit was attributed to the 1873 CE earthquake deposit.

[Figure]

**Figure 4**

*Correlation diagram for all cores in Lower Squaw Lake and relationship to the Upper Squaw Lake core.* Cores are hung on the lake-wide disturbance deposit J, suggested in the companion manuscript (Morey et al., submitted, this volume) to be the result of the 1700 CE Cascadia earthquake. The thick line connects deposits that are the result of a disturbance from around ~1500 BP.

[Figure]

**Figure 5**

Changes in expression of earthquake deposits downcore in SQB2. Archival depths are in cm below the core top. Deposits identified by red letters are disturbance deposits that are evaluated in this manuscript. Those identified by numbered boxes are illustrate the complexity and variability in the expression of these disturbance deposits downcore.

[Figure]

**Figure 6**

Identification of disturbance deposits and correlation between upper and lower lake records. A. Red numbers represent the interevent thicknesses used in the event-free age-depth model. The red capital letters A-Z indicate the disturbances identified in this study. Gray traces to the right identify correlative sequences where age data have been used to supplement those radiocarbon determinations in core SQB1/2/ss. B. The relationship to the 2009 Upper Squaw Lake core is shown by correlation lines (dashed). C. The relationship between the CT density data from core SQB1/2/ss (black trace) is shown compared to the CT density data from the upper lake core (green trace; 9 point Gaussian smoothing is shown over original data in blue). The relationships to the seven thickest deposits in the upper lake record compared to the lower lake record identified in Colombaroli et al., 2018 are identified by the dashed lines connecting numbers to events in the sequence. Note that the depth scale for the USL core (CT units shown in blue) are true, but the depth of the lower lake core (CT units shown in black) are not shown

because depths have been distorted to match events. This is called flattening. Breaks in the lower lake CT data were made in the middle of each thick deposit because the thicknesses of the upper lake deposits are much greater than the thicknesses of the lower lake deposits. Note that ages with +/- are radiocarbon determinations, and those with ranges in parentheses are calendar ages. See the Explanation for details.

d. *USL 2009 (left) was flattened to core SQB1/2/ss (right) to demonstrate the similarities between the core data*. Flattening is a method whereby all the core data are transformed to match correlative horizons, in this case, correlative deposit bases. Correlated bases are identified by the red tie lines between cores. The correlation suggests that the radiocarbon ages identified in gray are older than the radiocarbon data would suggest for the lower lake core. Note that whole round magnetic susceptibility is in black and CT density is in blue (for core USL 2009) and CT density is in black for the core SQB1/2/ss. The gray trace to the far right is the USL 2009 smoothed CT density (9 point Gaussian window) to better compare the records (because the data in the upper lake core contains many more silt layers than the lower lake core).

[Figure]

**Figure 7**

*Downcore age-depth model for Lower Squaw Lake composite core SQBss/1/2.* Sample numbers (refer back to Table 2) are positioned adjacent to their distributions. The envelope reflects the uncertainty (95% confidence) of the age-depth curve. Calendar ages in black are modeled ages and those in green are modeled ages by radiocarbon samples in close proximity to a disturbance deposit.

[Figure]

**Figure 8**

*Comparison of the Lower Squaw Lake earthquake chronology to the compilation of southern Cascadia paleoseismic records by Milker et al., 2016.* At the far left are the marine age ranges of corrected turbidite, margin-wide averages (corrected for reservoir age) from Goldfinger 2012. At the far right are the disturbance deposits distributions for deposits K, N, O, R, S, W and X which are most similar to deposit J. Those distributions in green are deposits that have been directly dated. The other distributions in lighter gray are the remaining disturbances in the sequence that have other characteristics (schist layers and thinner, less distinct, layers).

[Figure]

**Figure 9**

*Correlation diagram.* This diagram shows bed relationships for correlative units between Lower Squaw Lake, Rogue Apron and Hydrate Ridge Basin West paleoseismic sites.

[Figure]

**Figure 10**

*Correlated disturbance deposits in lake sediments near the California/Oregon border* (Figure 1 map for the locations of the lakes). Disturbance event deposits are shown as increases in CT density, magnetic susceptibility and loss on ignition data). Physical property signatures and radiocarbon age data allow beds to be correlated regionally. T1-T3 identify inferred relationships with marine sediment core events

from Goldfinger et al., 2012. Solid red lines identify the most confident ties between cores, and less-certain where dashed.

**Supplementary Files**

This is a list of supplementary files associated with this preprint. Click to download.

- Supplementarydata.docx

---

## Author Response (AR2)

Thank you to the anonymous reviewer for their comments! Please see responses in bold text below:
Author FINAL comments in red.

Reviewer 1: A 2700-yr record of Cascadia megathrust and crustal/ slab earthquakes from Upper and Lower Squaw Lakes, Oregon

By Ann Morey and Chris Goldfinger

The manuscript presents important paleoseimic data from a lacustrine archive at the southern end of the Cascadia subduction zone, where records have been scarce compared to the northern part. It also contributes knowledge to the characterization of so-called (disturbance) event deposits in organic-rich lake sediments where the identification of events can be difficult due to the homogeneous nature of such deposits as well as other factors such as low sedimentation rates etc. The amount of data presented is sufficient to support the main messages of the paper.

Despite the clear scientific merit, the concept used to identify earthquake-induced deposits and how these are generated remains somewhat unclear. There are also weaknesses in the background information and the structure of the paper. I believe these can be addressed but require some revision. Here are my major concerns:

1) It looks like that there is a companion paper to this paper characterizing the 1700 deposit in greater detail. Even if references are made, I believe that in order for this paper to stand alone, it must include more background information. I am wondering if the authors have considered to merge both papers?

**RESPONSE: Merging both papers was considered prior to submission but rejected based on the extended and complex arguments made in the first paper describing the 1700 deposit and timing. It seemed too much for one paper to combine them into a single paper.** **The authors remain convinced that these papers should remain separate.**

2) The seismicity of the southern Cascadia subduction zone has to be mentioned. For example, it is important to know the estimated groundshaking at the lake site for the historic earthquake in 1873 as well as the 1700 earthquake (or other large megathrust earthquakes).

**RESPONSE: Agreed. This will be addressed in a revision.** **DONE**

3) More importantly, the geologic setting of the lake must be beter characterized to understand the concept that is used to distinguish between large megathrust and smaller earthquakes. This includes the geology and geomorphology in the watershed of the lake as well as lake basin geometry and any delta or landslide deposits within the lake that could be destabilized during ground-shaking. This information is crucial to understand the source of the sediments found in type 1 deposits in contrast to the source of mineralogies found in type II deposits.

**RESPONSE: This has been adequately addressed in the first of the companion papers, but I can see the need to include this information in this paper as well. We will add a section on this.** **Some information has been included, and a reference to the companion paper has been added here.**

4) While the two types of deposits are relatively well characterized, I think that the processes that lead to the formation of these two types of event deposits remain somewhat uncertain. For example, I am having difficulties understanding what is meant by the watershed sourced turbidites. Do these turbidites incorporate sediment that comes from steep slopes in the surrounding watershed destabilized during ground shaking and subsequently transported into the lake (similar to NZ lakes in Howarth et al. 2014)? Is the other type of turbidite generated from slope failures within the lake? Is the difference of the two types just an effect of the amount of groundshaking at the site? I think I am not clear how you can distinguish between the different earthquake sources.

**RESPONSE**: **We will clarify that the watershed sourced portion of the disturbance event deposits (inferred to result from subduction earthquakes) are not strictly turbidites, but rather are inferred to be the result of the release of sediment (watershed sourced) and water into the water column during shaking which settles onto the organic sediment below as shaking ceases. These deposits do not incorporate sediment that comes from steep slopes in the surrounding watershed destabilized during ground shaking. Yes, the other type of disturbance event deposit is generated from slope failures within the lake. I can see that this needs clarification in the text.**

**We have now modified the text appropriately.**

5) I think that it is not necessary to correlate physical proxies of other lake studies as shown in figure 9 and 10. I would merge these records with figure 8 or include a similar figure that just shows the correlation of the ages for those sites. I think this will also streamline the discussion section.

**RESPONSE**: **It is very important to show the similarities in the physical property data between the onshore and offshore data (figure 9) because it demonstrates the underlying cause of the physical property data for both types of records is likely the same. We therefore think it's necessary to keep figure 9 as a stand-alone figure. We could add age ranges to figure 8, but none of the other distributions have ages; in other words, adding ages to figure 8 may not add anything. Likewise, it seems important that other regional lake records contain disturbance event deposits at the same time as the deposits inferred to reflect subduction earthquakes as support of this inference. However, there are multiple factors that influence how disturbance event deposits are recorded in each lake setting. Because of this, the relationships between records is not as clear; it may be acceptable to remove figure 10.**

After some deliberation it was decided that retaining figure 10 was important because it demonstrates how each individual lake should be independently evaluated for paleoseismic evidence.

Minor comments:

**Abstract:**

I would avoid specific deposit names such as deposit J, turbidite T1, T2 etc. in the abstract and the introduction since they have not been introduced, yet. Consider some rephrasing and add a sentence on the methods used.

**RESPONSE**: **This makes sense and will be done.** **This has been done as best as could be.**

**Introduction:**

Maybe include short review on how earthquake induced event deposits look like and what other lake studies have found in the area. I am wondering about overlap with the referenced paper Morey et al. 2023. Can the two papers be merged?

**RESPONSE**: **This makes sense and will be done, however there are no other studies on earthquake induced event deposits in regional lakes to compare with (other than Morey et al., 2013). As for merging the two papers, there are complex discussions in the first of the companion papers that would make combining the two papers very difficult to follow.** **Merging will not be done; these are companion papers and reproducing all of Morey et al. 2023 (now 2024) would make this an unwieldy manuscript.**

**Setting:**

The first paragraph under "Methods" seems to be geologic setting. As mentioned above, this has to be

expanded significantly regarding seismicity of the area and geologic setting of the lake.

**RESPONSE: This makes sense and will be done. This was expanded to include some information about the setting.**

**Methods:**

The Title of the manuscript implies that there is also a record from Upper Squaw Lake. However, the methods only describe cores taken from Lower Squaw Lake. Has data from Upper Squaw Lake already been published?

**RESPONSE: Yes. We will make sure that this is clearly referenced in the paper. It is clearly referenced with respect to Figure 6.**

Mention also XRD measurements that are mentioned later in the text.

**RESPONSE: Will do. Referenced Morey et al., 2024 (this volume)**

I believe the last three paragraphs of the methods section can be shortened and possibly restructured.

**RESPONSE: Will do. This was attempted.**

**Inferred characteristics for earthquake types**

See my major comment 4. It is not clear what is meant by "Sediment sourced from watershed" and "Turbidite sourced from lake margin bedrock".

**RESPONSE: This will be clarified as previously mentioned. FIXED**

Yes, the structures at the base look like load structures. The question is, if these formed due to rapid sedimentation at the time of turbidite deposition or subsequent seismic loading. Maybe you can disucss later?

**RESPONSE: We will include a paragraph on this. It was unclear where to do this because there is no place where details of deposits are discussed.**

For both types I am missing a short statement about the lateral, lake basin wide distribution.

**RESPONSE: This is because the cores from the deep water have some coring deformation due to the contrasting sediment and are more complex as they are sensitive to even minor disturbances compared to the shallower water, lake margin cores. That said, a short statement about the lateral distribution can be made. This was difficult to do because of the coring deformation.**

The list of characteristics at the end of this section seems to be a repetition.

**RESPONSE: OK Tried to fix this.**

**Results**

3.1: I would avoid the term "schist layers" FIXED

It seems as if the last two paragraphs that talk about correlation to Upper Squaw Lake should be under 3.2. I think the results section in general could be restructured a bit.

**RESPONSE: OK** Attempted this

**Discussion**

The process of earthquake disturbance layers has to be described in more detail. Unclear as to what is meant.

"Physical property peaks" is a litle too vague. Fixed

I think the post-fire and flood-related erosional events can be shortened especially since fires are already excluded as a trigger in Upper Squaw Lake sediments. Fixed

I would leave section 4.2.2 and Figure 9 out. It is tricky to correlate selected physical proxies from two very different environments.. The authors think it is important to make this tie between marine and lake environments; this demonstrates the usefulness of the two types of deposits.

4.3: There are some references and terms in this section that don't seem related to the paper. Tried to fix.

Section 4.4. is based on figure 9 and 10 which I would leave out and just add the age distributions to figure 8. See my comment 5. **RESPONSE: See our previous comments on this topic. IMPORTANT TO LEAVE IN (see previous comment)**

From the manuscript it is not clear how the summary of ideal lake characteristics was established. Tried to fix this. Under Summary and Conclusions there is another paragraph that talks about the suitability of regional lakes.
Tried to fix.

**RESPONSE: We agree to these comments other than the concerns about removing figures 9 and 10; see previous responses.**

**References** not checked – sorry!

Figures: I am not sure if I have access to the highest resolution possible for these figures. Some seem fuzzy and are hard to read.

Figure 1: Could it be cut above northernmost location mentioned in text (I believe above ~45 deg N). Could you add a smaller overview figure showing the entire Cascadia subduction zone/ northwest Pacific and mark extent of southern Cascadia?

Figure 2: I have a hard time reading the core names but it looks like this study is mostly based on cores from the lake side and not from the deep basin? It is important to explain why those were chosen since deep basin cores would probably show a more complete record.

Figure 3: What is meant by calcium minerals? If $CaCO_3$ data exits from LOI it might be useful to plot here especially if it correlates with a specific source area for the turbidites. I think the figure can be simplified. Not all smear slide pictures and detailed descriptions are necessary.

Figure 4: very hard to see details. I suggest plotting only selected cores at a larger scale.

Figure 6: I don't think D is necessary. Maybe merge information with A, B and C and enlarge.

Figure 8: Is it possible to add ages of events from Figure 9 and 10 here? Also, can you indicate what archive each site represents (marsh record, lake, offshore, etc.)

**Thank you for your helpful comments!**

**Agnon Comments from PDF** (author responses are in red)

**Make the landslide clearer on Figure 2.** Done

**Garmin fish finder –** speed of sound 1500 m/s? DONE

FIXED: "This has the benefit of radiocarbon age control data but does not use gamma density or p-wave velocity."

Section 2.1 137Cs data - Please consider pushing this part of the paragraph below the next one

"deposits attributed to earthquakes have organic rich tail deposits" FIXED

"It was inferred that the processes triggered by the earthquake" "It *is*Unclear what this refers to.